# Autophagy: A Key Regulator of Homeostasis and Disease: An Overview of Molecular Mechanisms and Modulators

**DOI:** 10.3390/cells11152262

**Published:** 2022-07-22

**Authors:** Laura Gómez-Virgilio, Maria-del-Carmen Silva-Lucero, Diego-Salvador Flores-Morelos, Jazmin Gallardo-Nieto, Gustavo Lopez-Toledo, Arminda-Mercedes Abarca-Fernandez, Ana-Elvira Zacapala-Gómez, José Luna-Muñoz, Francisco Montiel-Sosa, Luis O. Soto-Rojas, Mar Pacheco-Herrero, Maria-del-Carmen Cardenas-Aguayo

**Affiliations:** 1Laboratory of Cellular Reprogramming, Departamento de Fisiología, Facultad de Medicina, Universidad Nacional Autonoma de Mexico, Mexico City 04510, Mexico; jalim166@gmail.com (L.G.-V.); carmenaguila10@hotmail.com (M.-d.-C.S.-L.); salvadorfloresmorelos1998@gmail.com (D.-S.F.-M.); jazmingallardonieto@gmail.com (J.G.-N.); gus_lt2@hotmail.com (G.L.-T.); 201700001@estudiantes.upqroo.edu.mx (A.-M.A.-F.); 2Laboratorio de Biomedicina Molecular, Facultad de Ciencias Químico-Biológicas, Universidad Autónoma de Guerrero, Chilpancingo de los Bravo 39070, Guerrero, Mexico; zak_ana@yahoo.com.mx; 3Biotechnology Engeniering, Universidad Politécnica de Quintana Roo, Cancún 77500, Quintana Roo, Mexico; 4National Dementia BioBank, Ciencias Biológicas, Facultad de Estudios Superiores Cuautitlán, Universidad Nacional Autónoma de México, Cuautitlan Izcalli 53150, Estado de México, Mexico; jluna_tau67@comunidad.unam.mx (J.L.-M.); fmontiel_sosa@yahoo.com.mx (F.M.-S.); 5Banco Nacional de Cerebros-UNPHU, Universidad Nacional Pedro Henríquez Ureña, Santo Domingo 11805, Dominican Republic; 6Laboratorio de Patogénesis Molecular, Laboratorio 4, Edificio A4, Carrera Médico Cirujano, Facultad de Estudios Superiores Iztacala, Universidad Nacional Autónoma de México, Tlalnepantla 54090, Estado de México, Mexico; oskarsoto123@unam.mx; 7Red MEDICI, Carrera Médico Cirujano, Facultad de Estudios Superiores Iztacala, Universidad Nacional Autónoma de México, Tlalnepantla 54090, Estado de México, Mexico; 8Neuroscience Research Laboratory, Faculty of Health Sciences, Pontificia Universidad Católica Madre y Maestra, Santiago de los Caballeros 51000, Dominican Republic; mpacheco@pucmm.edu.do

**Keywords:** autophagy, molecular mechanisms, modulators, development, morphogenesis, diseases

## Abstract

Autophagy is a highly conserved lysosomal degradation pathway active at basal levels in all cells. However, under stress conditions, such as a lack of nutrients or trophic factors, it works as a survival mechanism that allows the generation of metabolic precursors for the proper functioning of the cells until the nutrients are available. Neurons, as post-mitotic cells, depend largely on autophagy to maintain cell homeostasis to get rid of damaged and/or old organelles and misfolded or aggregated proteins. Therefore, the dysfunction of this process contributes to the pathologies of many human diseases. Furthermore, autophagy is highly active during differentiation and development. In this review, we describe the current knowledge of the different pathways, molecular mechanisms, factors that induce it, and the regulation of mammalian autophagy. We also discuss its relevant role in development and disease. Finally, here we summarize several investigations demonstrating that autophagic abnormalities have been considered the underlying reasons for many human diseases, including liver disease, cardiovascular, cerebrovascular diseases, neurodegenerative diseases, neoplastic diseases, cancers, and, more recently, infectious diseases, such as SARS-CoV-2 caused COVID-19 disease.

## 1. Introduction

Autophagy (from the Greek words *auto*, meaning self, and *phagy*, meaning eating) is an important pathway that regulates the homeostasis of organisms; when this equilibrium is impaired, pathological conditions may develop. Autophagy also plays a housekeeping role in removing misfolded or aggregated proteins, clearing damaged organelles, such as mitochondria, endoplasmic reticulum, and peroxisomes, and eliminating intracellular pathogens. Furthermore, this pathway recycles primary resources for cells in stressful conditions, such as starvation. Once the autophagosome forms, it fuses the lysosome to degrade the cargo. The autophagosome fusion to the lysosome is called autophagy flux and is affected during neurodegenerative diseases. Autophagy plays a fundamental role in cellular, tissue, and organismal homeostasis and is mediated by conserved autophagy-related ATG genes. ATG genes have diverse physiologically essential functions in membrane trafficking and signaling pathways. Autophagy maintains cell homeostasis throughout its physiological role of recycling essential components for the cell and degrading potential harmful protein aggregates or microorganisms, functioning as a survival mechanism. Nevertheless, when it does not function properly or when its regulatory pathways are impaired, it could contribute to the development of several diseases. Therefore, the present review article aims to overview the physiological autophagy pathways and modulators and their implications in cell survival and development, and discuss the possible contribution to pathological conditions when impaired.

## 2. Autophagy

Autophagy is a lysosomal degradative intracellular process. This process allows the recycling of damaged organelles and aggregated or misfolded macromolecules. It is a survival mechanism that helps maintain cell homeostasis since it acts under stress conditions by recycling metabolic precursors and cleaning cellular debris. [1,2,3,4].

Autophagy helps to maintain cellular homeostasis in different conditions; for example, during nutritional stress, it allows the generation of energy or macromolecules; in cell differentiation, it provides precursor metabolites for morphogenesis; and when a molecule or organelle is no longer required or does not perform its function correctly, it degrades it to allow new synthesis to replace it. Table 1 summarizes some of the relevant autophagy functions.

### 2.1. Summary of Key Autophagy Players

As key autophagy players, we can identify several proteins among them as the ATG family (which will be described in detail in the following sections of this review). Over 32 related proteins have been identified that mediate the completion of the double-membrane autophagosome before fusion to the lysosome and the degradation of its contents. The formation of the autophagosome is initiated by a protein complex, which includes Ulk1 (Atg1) and Atg13, which in mammals can only be activated in the absence of signaling from the nutrient-sensing kinase mammalian target of rapamycin (mTOR). Additionally, the key to autophagosome development in humans is class III phosphoinositide 3-kinases Vps34 and Vps35, which, along with Beclin-1 (human homolog of Atg6), mediate the “nucleation” step. Elongation of the isolation membrane is mediated by two ubiquitin-like conjugation systems that are key to the autophagosome expansion: one system where Atg7 (an E1-like protein) and Atg10 (E2-like) act to conjugate Atg5–Atg12, and another where the Atg5–Atg12 conjugate (an E3-like protein) acts in concert with Atg7 and Atg3 (E2-like) to conjugate Atg8 to phosphatidylethanolamine, PE (so-called lipidation” of (LC3) for insertion into the membranes of the growing autophagosome [17,18,19]. LC3-I is uniformly distributed in the cell when autophagy levels are low, whereas upon induction of autophagy, lipidation of LC3 causes its relocalization to the autophagosome, which can be visualized and quantified by counting LC3 spots using immunofluorescence microscopy or by its differential migration in SDS-PAGE gels by Western blotting [20]. P62, also called sequestosome 1 (p62/SQSTM1), is a ubiquitin-binding scaffold protein, and is a classical selective autophagy receptor, but it also has roles in the ubiquitin-proteasome system, cellular metabolism, signaling, and apoptosis. Furthermore, p62 colocalizes with ubiquitinated protein aggregates in many neurodegenerative diseases and proteinopathies of the liver [21].

We will now describe the types of autophagy.

### 2.2. Types of Autophagy

Several types of autophagy are described below (Figure 1a and Table 2).

• **Macroautophagy** is the most common and studied form of autophagy. Macroautophagy is a process that leads to the bulk degradation of subcellular constituents by producing autophagosomes and autolysosomes. This mechanism implies that an isolating membrane (phagophore) sequesters a portion of the cytoplasmic substrates that degrade. The structure generated is called an autophagosome. Later, the outer membrane fuses with the lysosomal membrane; this allows for the combination of its content into the lysosomal lumen and gives rise to an autolysosome. In autolysosomes, degradation occurs by enzymatic action at acidic pH [1,17,22,23]. Some macroautophagy substrates include unnecessary and damaged organelles, cytosolic proteins, and invasive microbes. Autophagy is highly conserved from yeast to mammals [24]. *Atg5* and *Atg7* are considered essential genes for mammalian macroautophagy [25,26].

• **Microautophagy** is a process in which the cytoplasmic components enter the lysosome through invaginations or deformations in the lysosomal membrane [4,27,28]. This process is described in yeasts and preserved even in mammals [29]. Microautophagy degrades cellular components, including peroxisomes, mitochondria, lipids, and the nucleus, through selective and non-selective mechanisms [30,31].

Microautophagy falls into four sequential stages:Microautophagic invagination and autophagic tubes

In the early stage of microautophagy, the membrane bulges into the surface of the lysosome/vacuole. While certain lipids and lipid-modifying proteins drive and maintain a spontaneous pit, Vps1p regulates microautophagic invagination. The invagination extends into a termed ‘‘autophagic tube’’ [32,33].

2.Vesicle formation and expansion

Vesicle formation requires a sorting mechanism on the autophagic tube, lipid enrichment, and phase separation induced by integral protein depletion [33]. In microautophagocytosis, the enzymes involved in vesicle formation bind to the inside of the nascent structure and locally increase the curvature of the cytoplasmic leaflet. Microautophagic vesicle formation is related more to the formation of multivesicular bodies; that is, the formation of internal endosomal vesicles enriched in a specialized lipid (lysobisphosphatidic acid) that plays an essential role in the budding of vesicles into the vacuolar lumen. Vesicle budding into the vacuolar lumen occurs independently of the vacuolar SNARE proteins Vam3p, Vam7p, Nyv1p, Sec18p, and Sec17p [33,34].

3.Vesicle scission

The vacuolar transporter chaperone (VTC) complex is a calmodulin (Cmd1p) target in microautophagy. The VTC complex is distributed within vacuolar membranes, showing enrichment in autophagic tubes. The evidence suggests that the VTC complex is necessary for the scission of microautophagic vesicles from these tubes. The deletion of the VTC genes showed minor effects on the frequency of autophagic tube formation upon starvation. This suggests that VTC proteins act in autophagic tube organization or vesicle scission [35].

4.Vesicle degradation and recycling.

After scission from the autophagic tubes, the released vesicles move around in the lumen. First, Atg15 breaks down the vesicle [36], and then Atg22 acts as a permease for recycling nutrients and energy [37].

The diversity of microautophagic processes raises the question of how many types of microautophagies exist. Classifications can be based on the fact that microautophagy is a unified form of membrane dynamics that enwraps and transports cytosolic components into the lumen of the lytic organelle. In this case, the classification considers the microautophagy site (endosomes versus lysosomes) or the morphology of uptake intermediates (membrane invaginations versus extensions) [22]. On the other hand, a different classification based on the molecular mechanism of uptake is proposed, considering the new information gathered. This classification distinguishes between two principal types of microautophagy: fission-type and fusion-type microautophagy [38].

Fission-type microautophagy occurs through the invagination and fission of endosomal or lysosomal membranes. Therefore, it requires an endosomal sorting complex for transport (ESCRT). It is achieved by binding to ubiquitylated cargo to ESCRTs, or by factors such as Nbr1, LC3, and Hsc70, which recognize cargo and recruit it for binding to ESCRTs or membranes [38].

Fusion-type microautophagy requires core autophagy machinery and SNAREs to generate sealing membrane structures. Sealing occurs by vertex fusion, where two apposed vesicles do not fuse at a single point but along their vertex ring [39].

• **Chaperone-mediated autophagy (CMA)** is a type of autophagy that does not include the reorganization of membranes and is highly specific. This is because substrate proteins contain a pentapeptide biochemically similar to the sequence KFERQ. KFERQ functions as a label recognized by Hsc70 (Heat shock protein 70 kDa) and other cytosolic chaperone proteins [28]. Several studies have described the CMA process in mammalian cells. Subsequently, other findings define the process in birds, fish, fruit flies, and lower species, such as *C. elegans.*

The basic process of CMA involves the following steps:Substrate Selectivity

The properties of the KFERQ motifs, rather than the amino acid sequence, determine whether a protein is a potential substrate for CMA. The KFERQ-like motif of the substrate protein (KFERQ is a lysosomal-targeted signal peptide). Most known CMA substrate proteins contain the KFERQ peptide sequence, and only protein substrates with this sequence can bind and interact with molecular chaperone specificity. Detailed information about this motif has been reported [40]. The sequence is centered on glutamine, and four amino acid compounds form four flanks, including an alkaline end (K, R), an acidic end (D, E), a large hydrophobic end (F, I, L, V), and a repeating basic or large hydrophobic end (K, R, F, I, L, V). Such sequences are present in 30% of cytoplasmic proteins, some of which may produce KFERQ-like sequences through posttranslational modifications, such as the deamidation of posttranslational proteins [41].

2.Recognition of Substrates

Other chaperones are involved in the recognition of substrates. These chaperones are heat shock protein 90 (Hsp90), heat shock protein 40 (Hsp40), Hsp70–Hsp90 organizing protein (Hop), Hsp70-interacting protein (Hip), and bcl2-associated athanogene 1 protein (BAG-1), together with Hsc70, which form a chaperone complex that assists Hsc70 in recognizing and binding the KFERQ motifs of the CMA substrates [42].

3.Unfolding of the Substrate.

Some findings show that the folding state of a substrate is affected only by the translocation process. Therefore, it allows us to infer that the CMA substrate is in an unfolded state when it passes through the lysosomal membrane [43].

4.Substrate Translocation

LAMP2A acts as a transport channel in CMA substrate translocation. First, Hsc70 and the chaperone–substrate complex bind to the C-terminal tail of LAMP2A. Afterward, the substrate in the unfolded state translocates into the lysosomal lumen [44,45].

5.Degradation of Substrate

In the lysosomal lumen, lysosomal hydrolases degrade the substrate [46] while LAMP-2A disassembles from the multimeric complex. In this process, the disassembly of LAMP-2A requires hsc70 [45]. In addition, the association with the membrane lipid microdomain regulates LAMP-2A at the lysosomal membrane [47].

The diverse nature of CMA substrates justifies the variety of cellular processes impacted by CMA. For example, CMA is the first line of defense against protein aggregation by mediating the degradation of single proteins upon damage or partial unfolding (protein quality control) [48]. On the other hand, after at least 8 h of starvation, the CMA provides cells with free amino acids for energy generation and the synthesis of proteins required for cell function [49]. Moreover, the selectivity of CMA makes it suitable for some physiological functions: contributes to the regulation of glucose and lipid metabolism through selective degradation of key enzymes [50], it is essential for CD4+ T cell activation because it degrades two negative modulators of TCR signaling [51] and finally, is required to initiate cell cycle progression after DNA repair has been completed [52]. Basal CMA activity is detectable in almost all mammalian cells [53]. However, starvation, oxidative stress, genotoxic insults, lipid challenges, hypoxia, and radiation upregulate it [49,54,55]. Regarding CMA physiological regulation, cues from cytosolic signaling pathways and signaling events at the lysosomal membrane regulate the CMA. The first signaling pathway identified in CMA activation is the calcineurin–NFAT pathway. CMA activation in T cells requires this pathway, where the transcription factor NFAT1 binds to the *lamp2* proximal promoter region with a subsequent increase in its expression [51]. Physiological regulation of CMA activity also occurs at the lysosomal membrane through the TORC2–AKT1–PHLPP1 axis. PHLPP1 dephosphorylates AKT1, and inactivation of this kinase allows dephosphorylated GFAP to accelerate LAMP2A assembly and disassembly. These signaling events in the lysosomal membrane favor CMA fine-tuning to adjust to cellular needs [56].

In ordinary settings, macroautophagy and CMA occur at the basal level, but they can be activated when cells encounter stressful stimuli. For example, macroautophagy activity during nutrient deprivation reaches the maximum level around 4–6 h post fasting [57]. If nutritional stress continues, the CMA pathway upregulates. The pathway activity peaks at 12–20 h and remains active in the long-term starvation [49]. On the other hand, CMA-defective cells maintain protein degradation and cell viability through compensatory macroautophagy [58]. However, upon exposure to stressors, CMA-defective cells undergo more apoptotic cell death [59]. Intriguingly, the impairment of macroautophagy can lead to the activation of the CMA pathway [60]. However, under macroautophagy inhibition, CMA pathway enhancement fails to prevent menadione-induced cell death [61]. In summary, macroautophagy can degrade CMA substrates; however, it cannot compensate for the selective degradation of particular substrates by CMA. Similarly, CMA can degrade cytosolic proteins but fails to manage the macroautophagy-mediated degradation of expired organelles. In aging, there is reduced protein quality control in neurons [62], and defective autophagy is associated with neurodegenerative diseases, including Parkinson’s disease (PD) and Alzheimer’s disease (AD) [42]. In this sense, the CMA contributes to the degradation of pathogenic proteins involved in neurodegenerative disorders, such as α-synuclein (α-syn) [63] or tau [64]. Moreover, evidence suggests CMA deficiency in the aging brain is an aggravating factor in the onset of neurodegenerative diseases [65], even though inhibition of the neuronal CMA network is an early event in AD patient brain [66]. For this reason, it could be of interest to target CMA as a therapeutic strategy.

• **Mitophagy:** Mitochondria are essential energy organelles whose function regulation is critical for cell homeostasis. Many factors can affect mitochondria and consequently induce apoptotic cell death. However, cells have mechanisms to prevent this damage and specifically degrade dysfunctional mitochondria called mitophagy [17].

There are three types of mitophagies. Type 1 occurs when nutrients are scarce, but mitochondria are still functional or are in excess in the case of energy surplus. For this process, PI3K, Beclin-1, ULK1, and other proteins are necessary for phagophore formation and LC3 recruitment. During this process, mitochondrial fission and depolarization can occur. Finally, mitochondria degrade into lysosomes. Type 2, despite type 1, occurs when mitochondria suffer photodamage or another kind of damage. The process of type 2 is similar to type 1, except in the shape of phagophore formation, which forms a complete ring around damaged mitochondria. Before type 2, damage and depolarization of mitochondria should occur. Type 2 needs PINK1, Parkin, autophagy receptor protein like p62, and some autophagy receptors, such as Nix and Bnip3, to promote LC3 binding, and does not need PI3K, as a difference from type 1. Type 3 mitophagy is related to mitochondria-related vesicles (MDVs) formation and is called micromitophagy. Type 3 does not require mitochondria depolarization. MDVs become internalized into multivesicular bodies in a PINK1/Parkin-dependent pathway. MDVs fuse to the lysosome to complete the degradation of mitochondrial fragments. The generation of oxidative stress within mitochondria triggers type 3 mitophagy, which differs from canonical mitophagy. Therefore, type 3 works as a quality control for the remotion of damage and oxidized mitochondrial components. The differences in mitophagy types reflect different survival strategies for controlling cell homeostasis against harmful conditions and nutrient deprivation [3,67,68].

Furthermore, mitochondria are fundamental cell organelles because they provide energy and regulate a wide range of functions, including signaling and cell death (Figure 1b). The generation of reactive oxygen species (ROS) is correlated with the activity of the mitochondrial electron transport chain. Pathology may arise as a result of mutations, exposure to environmental contaminants, or long-term ischemic circumstances that disrupt mitochondrial redox equilibrium. In long-living, non-mitotic cells, such as neurons, oxidative stress brought on by an excess of mitochondrial ROS or a weakened antioxidant defense causes mitochondrial malfunction and initiates the cell death cascade. Numerous neurological illnesses and the majority of neurodegenerative diseases have been related to abnormalities in mitochondrial redox equilibrium and excessive generation of ROS [69], particularly oxitadive stress and mitochondrial dysfunction have been linked to AD and Parkinson’s disease PD [70], thus Among other factors that can induce oxidative stress are: Amyloid beta peptide (Aβ), pathogenic tau, α-synuclein, FUS, TDP-43, mutant SOD1, and mutant Huntingtin (Figure 1b).

• **Golgiphagy.** Recently, a new type of autophagy has been discovered, Golgiphagy [71,72,73,74], which modulates the amount of Golgi membranes. According to Rahman et al. (2022) [75], Golgiphagy is a selective type of autophagy that regulates Golgi Complex Turnover. The Golgi complex is formed by flat membrane sacks located around the cell nucleus, and its principal function is to prepare proteins and fat molecules for transportation and use in other places inside and outside the cell. Selective autophagy receptors and adapters contain short linear motifs called LIR motifs (LC3-interacting regions), which are required for interaction with Atg8-family proteins. LIR motifs bind to the hydrophobic pockets of the LIR motif docking site (LDS) of the respective Atg8-family proteins. For this selective autophagy process, Atg8a’s LDS docking site is important. The fact that the GMAP (Golgi microtubule-associated protein) protein interacts with Atg8a and the LIR motif at position 320–325 is important for the docking between the phagophore and the Golgi complex. Therefore, the role of this type of selective autophagy is to regulate the size and morphology of the Golgi apparatus. Thus, Golgi complex stress has been linked to the development of aging-related diseases; this mechanism of autophagy may be crucial in maintaining cell homeostasis.

### 2.3. Mitochondrial Stress and Autophagy

Mitochondria are the principal source of Reactive Oxygen Species and Reactive Nitrogen Species (ROS/RNS) [84,85] for autophagy induction [86]. ROS/RNS regulates autophagy at different levels and through multiple mechanisms. For instance, in response to H_2_O_2_, caveolin 1 promotes the translocation of BECN1 to mitochondria and facilitates autophagosome formation [87]. In addition, there is a relationship between ROS and Ca^2+^ signaling in regulating autophagy through transcription factor EB (TFEB) activation (Figure 1b) [88].

ROS can also regulate autophagy through the pathway of mitogen-activated protein kinase (MAPK). The MAPK pathway affects the activity of the transcription factor activator protein 1 (AP-1), forkhead box transcription factor O (FoxO), and nuclear factor kappa B (NF-kappa B), which regulate autophagy-related gene expression [89,90]. On the other hand, ROS can also induce autophagy by regulating mTOR [91]. Moreover, moderate levels of ROS can lead to JNK pathway activation and increase autophagy levels. However, a certain level of ROS leads to continuous activation of JNK and induces apoptosis mediated by mitochondrial pathways [92]. Finally, redox processes mediate changes in ATG4, thereby regulating the formation of autophagosomes [93,94].

In autophagic signaling, nitric oxide (NO) and RNS have different effects depending on the cell type. For instance, breast cancer cells, such as MCF-7, are sensitive to the induction of autophagy by nitrosative stress. In these cells, the evidence indicates that the activation of ATM/LKB1/AMPK/TSC2 signaling cascade by nitrosative stress represses mTORC1 and induces autophagy [95]. Under RNS, in the same cells, another study shows the autophagy induction by SIRT1, phospho-AMPK, and p53 [96]. In the K562 leukemic cell line, nitric oxide treatment also induces autophagy. The exposure transmits signals to AMPK to phosphorylate it, and phosphorylated AMPK induces autophagy [97].

An exacerbated ROS/RNS generation by dysfunctional mitochondria can shift its role from an autophagy inducer into a self-removal signal for mitochondria through mitophagy, a delicate mechanism of regulation by which autophagy eliminates the source of oxidative stress and protects the cell from oxidative damage [98,99]. Thus, impaired mitophagy could underly many pathological conditions, including neurodegenerative diseases, cancer, and aging (Figure 1b) [100].

Post-mitotic neuronal cells are more susceptible to mitochondrial damage due to their increased metabolic demands [101]. On the other hand, oxidative stress and mitochondrial dysfunction are associated with the aggregation of misfolded proteins in neurodegenerative diseases, such as AD, PD, and Huntington’s disease [102,103]. In line with this, the brains of individuals with these diseases also display an accumulation of autophagosomes. Furthermore, the above suggests impaired autophagic flux as the cause of abnormal disease-specific protein buildups [104,105]. In vivo studies in mouse models have shown that autophagy inhibition enhances degeneration symptoms [106].

Conversely, pharmacological interventions on autophagy/mitophagy pathways allow the clearance of neurotoxic aggregates. The presence of misfolded proteins (Aβ, pathogenic tau, α-synuclein, Fus, TDP-43, mutant SOD-1, and mutant Huntingtin) can also affect mitochondria through damage to mitochondrial DNA, alteration in their trafficking and dynamics biogenesis, deregulation of bioenergetics and quality control pathways, or promoting mitochondrial-dependent cell death pathways [107,108]. Whether mitochondrial dysfunction is a cause or consequence of neurodegenerative disease pathogenesis is still unknown [109]. However, a self-perpetuating feed-forward toxic cycle may exist since it is unclear which process is first: the proteinopathy (aggregation of misfolded proteins) causes mitochondrial damage, or vice versa.

### 2.4. Proteins Involved in the Autophagy Pathway

Autophagy is a process that comprises several stages (induction, nucleation, elongation, fusion, and degradation) regulated by proteins belonging to the ATG family (Autophagy Related Genes). More than 35 genes code for this family of proteins in yeast and fungi. However, homologous proteins in plants and higher mammals have been identified (Table 3 and Figure 2), emphasizing that autophagy is a conserved pathway [110,111,112].

### 2.5. Description of Autophagy Proteins According to Their Participation in the Autophagy Stages

#### 2.5.1. Induction

ATG1

Atg1 is a conserved serine-threonine kinase required for both selective and bulk autophagy pathways [115]. Atg1 consists of 897 amino acids and possesses two globular domains: the serine/threonine kinase domain (KD) at the N-terminal region and two tandem microtubule-interacting and transport (MIT) domains at the C-terminal region [116]. Atg1 forms a pentameric complex with Atg13, Atg17, Atg29, and Atg31 [117,118], whereas mammalian ULK1 forms a tetrameric complex with Atg13, Atg101, and FIP200, also known as the ULK1 complex [119,120].

ATG13

Atg13 is an adaptor protein that TORC1 and PKA hyperphosphorylate under normal growth conditions [121]. Atg13 consists of two distinct domains: the N-terminal HORMA (from Hop1, Rev7, and Mad2) and the C-terminal disordered region. The C-terminal disordered region is responsible for binding to Atg1 and Atg17, which are required to assemble the ULK1 complex [122]. Mechanistically, the nutrient-regulated phosphorylation of ATG13 controls the induction of autophagy by modulating the distinct properties of ULK1, including enzyme activity and cellular localization [123].

ATG17

Atg17 is a scaffold protein that forms a complex with Atg1-Atg13 under the control of Tor. This complex is required for proper autophagosome formation directed by Atg1 kinase activity [124]. Atg17 interacts with both Atg1 and Atg13 via two coiled-coil domains, and these interactions facilitate its inclusion in the Atg1 complex [125].

#### 2.5.2. Nucleation

ATG6

Atg6/Beclin 1 is an evolutionarily conserved protein family that has been shown to function in autophagy in a wide range of species [126,127]. In mammalian cells, Beclin 1 binds to Vps34, and Beclin 1-associated Vps34 kinase is activated explicitly upon amino acid starvation [128]. Beclin 1 contains three structural domains: a BH3 domain (amino acids 114–123) at the N-terminus, a central coiled-coil domain (CCD, amino acids 144–269), and an evolutionarily conserved domain (ECD, amino acids 244–337) [129]. ECD is essential for Vps34 binding, autophagy, and tumor suppressor function [130].

ATG9

Atg9 is a transmembrane protein that possesses six transmembrane helices [131]. Atg9 is incorporated into a vesicle termed an Atg9 vesicle at the Golgi apparatus, which is recruited to the pre-autophagosomal structure (PAS) upon starvation and functions as an initial membrane source for the isolation membrane (IM) [132]. In addition, Atg9 has some lipid-translocating activities and collaborates with Atg2 for IM expansion [133].

ATG14

Atg14 is a membrane tether [134]. It promotes and stabilizes the STX17–SNAP29 binary t-SNARE complex assembly in autophagosomes and primes it for interaction with endolysosome-localized VAMP8 to promote membrane fusion between autophagosomes and endolysosomes [135].

ATG18

In mammals, WIPI1 and WIPI2 are homologous to Atg18. Atg18 binds to PI(3)P and PI(3,5)P2 via two sites located on Blade 5 and Blade 6, as well as on the 6CD loop [136]. Interaction of Atg18 with these phosphoinositides is essential for its localization to the PAS, endosomes, and vacuole [137]. Atg18 forms a complex with Atg2. PtdIns(3)P-binding of Atg18 is involved in directing the Atg18-Atg2 complex to autophagic membranes [138].

#### 2.5.3. Elongation

ATG3

In autophagy, Atg3 contributes to autophagosome formation by interacting with Atg7, Atg8, Atg12, and the lipid membrane [139]. The structure of Atg3 is composed of a head region (core region) and a handle region (HR). The core region is similar to canonical E2 enzymes. The HR consists of a long α-helix and a partially disordered loop region. Furthermore, Atg3 features a “floating” helix C called FR at the interface between the core region and the HR [140].

ATG4

Atg4 is a cysteine protease and plays a central role in the Atg8/LC3 lipid conjugation system, which is essential for the late step of autophagosome formation, i.e., the expansion and closure of the membrane [141]. Atg4 has a well-conserved region that corresponds to the region from amino acids 154–172 and includes the conserved cysteine residue, which is probably part of the active site [142]. During autophagy, Atg4 removes the C-terminal arginine of Atg8, which is a crucial step required for Atg8 to covalently link to phosphatidylethanolamine (PE) on isolation membranes [143].

ATG5

Atg5 is a protein involved in the early stages of autophagosome formation [19]. Atg5 binds with Atg12 and is catalyzed by Atg7 and Atg10. Atg12 C-terminal glycine residue generates a covalent conjugation with a lysine residue of Atg5, forming the Atg5-Atg12 complex. After Atg5 binds covalently with Atg12, the complex interacts with Atg16 [144]. The Atg5-Atg12-Atg16 complex is a ubiquitin-like conjugation system that contributes to the elongation of the isolated membrane and autophagosome maturation [145].

ATG7

Atg7 is a common ubiquitin E1-like activating enzyme [146]. Atg7 is involved in two conjugation pathways: the conjugation of Atg8 to PE, and the conjugation of Atg12 to Atg5 [147] is indeed essential for LC3 modification systems and autophagosome formation [148].

ATG8

Atg8 and its mammalian homologs, LC3, and GABARAP proteins, have structural homology to ubiquitin. They are synthesized as precursors and undergo processing by Atg4 proteases. Mature Atg8 is activated by Atg7 (E1 enzyme), transferred to Atg3 (E2), and linked to the amino group of PE [149]. Mechanistically, ATG8 roles in autophagy can be attributed to their functions as binding platforms for other proteins, particularly once attached to the membrane [150]. The conjugation of Atg8 to phosphatidylethanolamine, PE (so-called lipidation” of LC3), is necessary for its insertion into the membranes of the growing autophagosome.

ATG10

Atg10 is an autophagic E2-like enzyme that interacts with Atg7 to receive Atg12 and participates in autophagosome formation [151]. Atg3 and Atg10 possess insertion regions in the middle of the core fold and may be involved in protein function. The missing segment termed the ‘FR-region’ in Atg10 may be necessary for interaction with the Atg7 E1 enzyme [152].

ATG12

Atg12 is a ubiquitin-like molecule activated by the E1-like enzyme Atg7, transferred to the E2-like conjugating enzyme Atg10, and ultimately attached to Atg5 [153]. However, the principal substrate of Atg12 is Atg5, Atg3 as an additional Atg12 target [154]. Atg12–Atg3 conjugation promotes autolysosome formation under nutrient-rich conditions [155].

ATG16

Atg16 is a scaffold protein that interacts with Atg12–Atg5 protein conjugates via its N-terminal domain and self-assembles via its coiled-coil domain (CCD), forming the Atg12–Atg5–Atg16 complex [156]. The protein comprises an N-terminal half with a CCD for homo-dimerization and binding sites for Atg5, clathrin, FIP200, and Rab33B [157,158] and a C-terminal β-propeller structure that binds NOD 1 and 2, TMEM59, and ubiquitin [159,160]. The N-terminal region and CCD are indispensable for canonical autophagy. The C-terminal WD40 repeat domain seems to have additional functions in bacterial sequestration and Crohn’s disease [161].

### 2.6. Autophagy Is a Multistep Pathway; The Following Are the Main Steps in the Autophagy Process

#### 2.6.1. Initiation Stage

Autophagy-inducing factors are responsible for the initiation stage. ULK1 protein, dephosphorylated and dissociated from the mTORC1 complex, and ATG13 and FIP200 phosphorylations, initiates the autophagy process [162]. ULK1 regulates two principal regulatory kinases: mammalian target of rapamycin (mTOR) and AMP-activated protein kinase (AMPK). Signals such as starvation suppress mTOR and activate AMPK, which results in ULK1 activation. Additionally, AMPK could directly inhibit mTOR. Subsequently, the ULK1 complex phosphorylates the Ambra1 protein and allows Beclin-1-Vps34 complex (Beclin1-Vps34-Vps15-ATG14L) recruitment in the autophagosome formation site. Then, Vps34 produces phosphatidylinositol-3-phosphate to recruit the effector protein DFCP1, which promotes the development of the double membrane during nucleation [28]. The phagophore is generated from the ER or Golgi membrane, mitochondria, and plasma membrane via endocytosis mediated by clathrin. In some autophagy subtypes, there are several autophagy receptors, p62 (SQSTM1), that mediate the clearance of ubiquitinated proteins (Figure 2a).

#### 2.6.2. Expansion (Elongation)

The ATG12-ATG5-ATG16L and phosphatidylethanolamine (PE)-LC3 systems are involved in the elongation of phagophores. These two interactors are ubiquitin-like conjugating systems [28]. By cleavage of pro-LC3 by ATG4B, the cytosolic form of LC3 (LC3-I) is generated. Then, ATG7 processes LC3-I and ATG3 to be conjugated to PE and form LC3-II. Therefore, LC3-II puncta visualized by immunofluorescence reflected the number of autophagosomes (Figure 2a).

### 2.7. Mechanisms of Autophagosome or Amphysome Fusion to the Lysosome

One relevant autophagy step is the fusion of Autophagosomes or Amphisomes to the Lysosome to allow autophagy flux. We will describe some of the fusion components and their mechanisms below (Figure 2b).

The formation of bundles from soluble N-ethylmaleimide-sensitive factor attachment protein receptor (SNARE) proteins localized in opposing membranes drives membrane fusion. Two SNARE complexes act to mediate the fusion of autophagosomes with late endosomes (LEs)/lysosomes. One complex consists of STX17-SNAP29-VAMP8 [163], while another is composed of YKT6-SNAP29-STX7 [164]. The autophagosome—LE/lysosome fusion process requires SNARE complex disassembly on post-fusion membranes. To enhance vesicle fusion, tether proteins are recruited. Some tether proteins are EPG5, homotypic fusion and protein sorting (HOPS) complex, PLEKHM1, baculovirus IAP repeat-containing ubiquitin-conjugating enzyme (BRUCE), and GRASP55. Another component that mediates autophagosome-lysosome fusion is Rab-7. On LEs/lysosomes, Rab7 and Mon1-Ccz1 (Monensin sensitivity protein 1—Caffeine, calcium, and zinc 1) complexes are located. In addition, Rab7 recruits tethering factors (EPG5, PLE KHM1) [165], and HOPS to promote SNARE complexes’ assembly for fusion. Atg8 proteins also contribute to tether recruitment and other proteins, but they must be removed to allow subsequent fusion [166]. The lipid composition of the lysosomal membrane also regulates the fusion capability of lysosomes. For instance, reduction or elevation of the lysosomal PI (3,5) P2 level impairs autophagosome/amphisome-lysosome fusion [167].

The next step is the fusion mechanism between the autophagosome and lysosome: Mon1-Ccz1 recruits and activates Rab7. Rab7 binds to FYCO1 (FYVE and coiled-coil domain-containing 1), ORP1L (oxysterol-binding protein-related protein 1L), and RILP (Rab-interacting lysosomal protein). Therefore, it can mediate the plus-end and minus-end transport of vesicles. The transport of autophagosomes and lysosomes toward each other is essential for fusion. Once the autophagosomes and lysosomes reach their destination, they tether to their partner organelle. Large coiled-coil proteins or multi-subunit tethering complexes, such as EPG5, HOPS complex, PLEKHM1, BRUCE, and GRASP55, carry out tethering, and they bind to Rab-7. Next, SM (Sec1/Munc-18) family proteins arrive and facilitate SNARE complex assembly and zippering. During vesicle fusion, an R SNARE located on the membrane of one vesicle and three different Q-SNAREs on the other form a trans-SNARE complex (STX17-SNAP29-VAMP8//YKT6-SNAP29-STX7). The zippering of these domains fuses the membranes, and SNAREs are located on the same membrane. This cis-SNARE complex is then recognized and disassembled by NSF (N-ethylmaleimide sensitive fusion protein) and a-SNAP (alpha soluble NSF-attachment protein) proteins for recycling.

### 2.8. Autophagosome Degradation and Recycling

Mature autophagosomes fuse with lysosomes to form autolysosomes (Figure 2a), where lysosomal enzymes digest cytosolic compounds. Finally, the degradation products are released into the cytosol for recycling in the synthesis of new macromolecules [162].

Once autophagosomes are formed, the dynein–dynactin complex transports them onto the endosomes and lysosomes [168]. Therefore, retrograde transport is crucial for the degradation of autophagosome cargo. In neurodegenerative diseases, cytoskeletal integrity is compromised, and vesicular transport is impaired. In some cases, autophagosomes fuse with late endosomes and then with the lysosome or directly with the lysosome. In lysosomes, degradation occurs with hydrolytic enzymes that are active in acidic pH. The vacuolar ATPase (vATPase) regulates pH, which imports hydrogen ions. vATPase induces transcription factor EB (TFEB) activity in starvation conditions where mTOR is inhibited, and the TFEB is phosphorylated. In these conditions, TFEB translocates to the nucleus and induces the transcription of genes that promote autophagy [168,169,170].

It is important to note that the measurement of levels of early or late autophagic compartments, or the autophagic flux, allows for monitoring autophagy in different organisms, according to the guidelines for standardizing research in autophagy published in 2021 [20] because there are no absolute criteria for determining autophagic status that are applicable in every experimental context.

## 3. Physiological Role of Autophagy during Morphogenesis

Systemic and tissue-specific ATG gene knockout mice analyses have rendered significant advances in our understanding of the roles of autophagy in mammalian development and differentiation. The autophagy pathway seems essential for two critical stages of early development: the pre-implantation period after oocyte fertilization and the early postnatal period after disruption of the placental food supply. Autophagy could also have a role in other stages of embryogenesis because the deletion of some ATG genes leads to lethality during mid-embryonic development, and the ATG-gene knockout mice that survive the postnatal period display some developmental abnormalities [18].

Morphogenesis is a set of biological phenomena that allows living beings to develop specific structures during each of its components’ formation, growth, and maturation. It involves morphogenetic mechanisms, such as cell differentiation, growth, migration, induction, cell death, and apoptosis.

During embryonic development, the cell must undergo numerous morphological changes that are decisive for the shape of the embryo’s body, initiating organogenesis and differentiation into functional tissues. These remodeling processes are accompanied by profound changes in the cell membrane, the cytoskeleton, the organelles, and the extracellular matrix composition [171].

Throughout cell differentiation, cells must degrade obsolete proteins to make room for the synthesis of new proteins. Autophagy and the ubiquitin-proteasome system (UPS) degrade these unnecessary protein complexes to produce substrates that will serve to synthesize new proteins necessary for cellular activities related to morphogenesis [171,172].

Several studies suggest that autophagy plays a role in development, but its specific functions remain unclear [172]. In mammals, fertilization induces high levels of autophagy, which plays an essential role in the early stages of embryogenesis. In particular, just after fertilization, where there is degradation from maternal proteins, messenger RNAs (mRNAs), and the synthesis of new proteins encoded by the zygote genome, autophagy allows nutrient supply for embryos until implantation due to the characteristic limited availability of nutrients in this particular period. In addition, autophagy in early embryos is essential for eliminating paternal mitochondria [17]. Moreover, it has been suggested that the induction of autophagy is required for the deep cytoplasmic remodeling process that cells face in specific stages of their development due to the rapid changes in ATG gene expression and assembly of protein complexes during embryonic development [171,172].

In mammalian systems, autophagy is thought to be involved in many physiological processes, including the response to starvation, cell growth control, and antiaging mechanisms. A role of autophagy in the regulation of vertebrate development has long been proposed by studies reporting morphological features of autophagy during embryogenesis. Targeted mutagenesis of a few ATG genes in mice substantiated this hypothesis [173]. The data suggest an essential role for autophagy genes in early development in Drosophila, *C. elegans*, and mice. Higher concentrations of RNAi directed against *C. elegans* orthologs of yeast ATG6, ATG8, and ATG18 result in early developmental arrest during or before the first larval stage. In mice, targeted disruption of Beclin-1 results in early embryonic lethality [174].

In the same way, apoptosis participates together with autophagy in the remodeling and elimination of numerous structures during embryonic development. For example, in human and other mammal limb development, the five individualized fingers in each limb prove the elimination of interdigital areas. In comparison, evidence shows that in the advanced stages of interdigital tissue, autophagy contributes to the regression of this tissue [175].

Apoptosis is a mechanism of programmed cell death essential for homeostasis and the development of living organisms. Cellular stress events promote the activation of proapoptotic signaling pathways that activate cysteine proteases called caspases and a complex cascade of events that cause mitochondrial dysfunction, causing apoptotic cells to experience changes in cell morphology, such as cell rounding, blebbing of the plasma membrane, and nuclear fragmentation [176]. Although autophagy is an intracellular catabolic mechanism that involves the degradation of the recycling of unwanted cytoplasmic components to maintain cell survival and homeostasis, it also cross-regulates apoptosis depending on cell type, intracellular metabolic activity, nutrient supply, and triggering stimuli [177]. Given the cytoprotective role of autophagy, it appears that the induction of apoptosis is coupled with the inactivation of autophagy. Some of the molecular events that show that autophagy and apoptosis are antagonistic events that tend to inhibit each other are the cleavage of Beclin-1 by Caspase-3, destroying its proautophagic activity. Beclin-1 cleavage fragment presents new functions by amplifying mitochondria-mediated apoptosis; likewise, the activation of Caspase-3 cleaves and activates Atg4d, which is responsible for the elimination of lipids associated with LC3 [178]. Moreover, the proteolytic activity of calpain cleaves Atg5, preventing its proautophagic function. On the other hand, autophagy is capable of mediating the intrinsic pathway of apoptosis initiated by mitochondrial membrane permeabilization if this event is limited to a fraction of mitochondria, resulting in the selective removal of depolarized mitochondria and the prevention of cell death. Furthermore, the release of Bcl-2 and FLIP from autophagy-activated protein complexes blocks apoptosis’s intrinsic and extrinsic pathways [178,179]. Therefore, autophagy is an adaptative response to stress conditions, allowing the survival of the cells for a longer time in adverse environments. Hence, autophagy and apoptosis are in some way mutually exclusive, but autophagy in the latter stages can converge into late apoptosis under harmful conditions (Figure 3) [180,181].

### 3.1. Autophagy as a Key Regulator of Embryonic Development

Embryogenesis in mammals is the process by which the embryo develops and starts with the division of one single cell into two-, four-, and eight-cell stages and the generation of morula and blastula structures. Embryonic development is characterized by the regulation of the mechanisms of proliferation, migration, differentiation, and organ development. These processes are controlled by key signaling pathways. Some of them are FGF, Hedgehog, Wnt, TGFβ, and Notch, pathways that operate during development, eliciting diverse cellular responses (Figure 4) [172,182,183,184,185].

In addition, cellular culture studies and mouse models suggest that autophagy is essential for development [186]. For example, the knockout model Atg5^−/−^ mice show embryonic lethality at the four-cell to the eight-cell stages [187]. Knockout models of Atg3 [188] and Atg7 [148] also cause neonatal lethality. The analysis of these models indicates that amino acid levels are decreased, suggesting that autophagy is necessary to maintain the amino acid pool during the early neonatal period. Interestingly, neuronal development is sensitive to autophagy. This statement is supported by the findings that activating molecules in Beclin 1-regulated autophagy protein 1 (Ambra1)^−/−^ mutants have embryonic lethality, with defects in the neural tube and hyperproliferation, followed by apoptotic cell death; thus, Ambra1 is an important factor at the crossroad between autophagy and apoptosis, controlling the balance and conversion between autophagy and apoptosis [189].

Some reports show that autophagy interacts with the developmental pathways of FGF, Hedgehog, Wnt, TGFβ, and Notch [15,190,191,192,193]. The impairments in the crosstalk between these pathways and autophagy could be a likely cause for the defects in development observed in the knockout of many autophagy-related genes in mice since the pathways are important for the formation of embryonic structures.

Regarding the Notch signaling pathway, in an ATG7 or ATG16L1 knockdown cell model, the depletion of the levels of these core autophagy proteins increased the levels of Notch1, the activated form of Notch (NICD), and Hes1. On the other hand, ATG7 or ATG16L1 overexpression, Beclin-1 overexpression, and autophagy stimuli (rapamycin or starvation) reduced the levels of Notch1, NICD, and Hes1. The results suggest that autophagy modulation alters Notch signaling. The findings of other studies indicate that Notch1, but not NICD or Hes1, is degraded by autophagy, demonstrating developmental retention of early-stage cells in various tissues where the differentiation of stem cells is delayed, revealing how modest changes in autophagy can impact stem cell fate, affecting neurogenesis, hematopoiesis, and gut villi in a mouse model with a hypomorphic mutation in Atg16L1 [15]. In *Drosophila* egg development, loss of autophagy leads to activation of the Notch pathway through downregulation of *Cut* and upregulation of *Hnt*, two Notch downstream target genes [194]. Finally, Notch and Wnt signaling pathways play important roles in heart development. In an in vitro cardiomyocyte differentiation system, the findings showed that β-Catenin and NICD, effectors of Wnt and Notch pathways, form a complex with LC3 and p62 and are degraded by autophagy. The clearance of these effectors facilitates the cardiac differentiation process (Figure 4) [195].

Moreover, a report indicates that autophagy attenuates Wnt signaling by enhancing Dvl2 degradation through p62 in cooperation with LC3 to mediate recruitment to autophagosomes, leading to its degradation [191]. Other signaling pathways related to autophagy and development include the Sonic hedgehog (Shh) signaling pathways. These pathways regulate osteoblast differentiation and morphological transition. The findings of a study showed that whereas autophagy inhibits osteoblast differentiation, activation of the Shh signaling pathway promotes osteoblast differentiation and development. Likewise, the crosstalk between autophagy and Hh signaling indicates that the Hh signaling pathway negatively regulates autophagy, inhibiting autophagy protein ATG5, which implies that autophagy might be a downstream pathway of Hh signaling [196]. Lastly, in the cytotrophoblasts (CTBs) of women with recurrent miscarriage (RM), Shh signaling was impaired in patients compared to normal controls, activating autophagy to inhibit trophoblast motility. The results showed that inhibition of Shh signaling promotes autophagosome maturation and decreases the expression of VEGF-A, suggesting poor vascular placentation in recurrent miscarriage patients (Figure 4) [197].

In alveolar morphogenesis, TGF-β1 participates in the morphogenesis and remodeling of the mammary gland through the induction of autophagy in a 3D model of bovine mammary epithelial cells (MECs) [198]. TGF-β1-induced autophagy showed increased levels of Berclin1 and LC3 proteins and ultrastructural features, such as autophagosomes and autophagic vacuoles (Figure 4) [199].

Finally, fibroblast growth factor (FGF) signaling is a regulator of autophagy in chondrocytes. In a Fgf18^+/−^ mice, the findings indicate that FGF18 induces autophagy through the regulation of VPS34–beclin-1 complex activity. The data suggest that chondrocyte autophagy regulates postnatal bone growth [200]. The results of another report show that ablation of the FGFR1/2-FRS2α signaling in heart progenitor cells led to premature differentiation. They demonstrated that FGF signaling suppressed cardiac cell differentiation by inhibiting autophagy due to the ablation of Frs2α increased the numbers of green fluorescent protein (GFP)-LC3-positive vacuoles, corresponding to an increase in the formation of autophagosomes (Figure 4) [201].

Considering all the above-mentioned evidence, we can consider that autophagy plays a crucial role in the regulation of embryonic development (Figure 4).

### 3.2. Physiological Role of Autophagy in the Development of the Central Nervous System

Many neural cells undergo programmed cell death (PCD) during embryonic development. This active elimination of cellular populations is a necessary process for the proper establishment and maintenance of the nervous system [202]. Neural cell PCD occurs during most stages of development, including proliferation, migration, axonal guidance, and synaptogenesis. This suggests PCD has three main functions during embryonic development: (i) limiting the size of the proliferation pool during neurogenesis because neurons and glia generate in large numbers; (ii) correcting errors and eliminating transitory structures of development, which refines the cytoarchitecture and connectivity of the nervous system [202] and (iii) quantitatively pair neurons with efferent targets and afferent inputs for optimal neuronal connections during synaptogenesis (Figure 5) [203].

The central nervous system (CNS) develops from the neural tube in the vertebrates, where neural stem cells can give rise to neurons, astrocytes, and oligodendrocytes [204]. During development, basal autophagy regulates Wnt and Notch signaling, which are required primarily for proper neuronal differentiation. For its part, Notch signaling is crucial from embryogenesis to adulthood, and it is a master regulator of neural stem cells and neuronal development, where it is often used to select from pre-existing developmental signals and anticipate decisions about cell fate [15]. The generation process of new neural precursor cells is called neurogenesis, and the trophic factors tightly regulate it (Figure 5 and Figure 6).

Studies show that autophagy participates in the early development of the nervous system, adequate membrane renewal at axon terminals, and neurogenesis [205]. Autophagic machinery interruption alters proliferation, differentiation, and cell death. These processes are necessary to establish the nervous system’s cytoarchitecture [202]. Vazquez’s lab demonstrated that the expression of ATG7, Beclin1, LC3, and Ambra1 increases markedly in neuronal stem cells (NPCs) derived from the olfactory bulb (OB) of embryonic mice in culture during the initial period of differentiation neuronal [206]. Ambra1 is a highly conserved protein in vertebrates that binds to Beclin1 during the nucleation process in autophagy. During CNS development, it controls cell proliferation and guarantees cell survival during this process [207]. Moreover, the report indicates that Ambra1 and ATG5 deficiency decrease neurogenesis [206], and Ambra1 homozygous mutations cause embryonic lethality with tube defects, and, neural disease such as midbrain/rhombencephalon exencephaly or spina bifida around day 14.

### 3.3. Effect of Autophagy on Neuronal Survival

Autophagy is critical in preserving the viability of neurons and glial cells because mature central nervous system (CNS) cells are post-mitotic. Therefore, they cannot simply discard dysfunctional cellular components through the process of cell division. For this reason, to maintain homeostasis and support long-term viability and functionality, they require robust quality control mechanisms. In addition, the maintenance of neuronal and glial activity imposes high energy demands that are resolved due to the recycling of essential components through autophagy. Consequently, autophagy is considered predominant for neuronal homeostasis and survival (Figure 3 and Figure 5) [23,208].

The loss of neurons is associated with increased ubiquitin-positive protein aggregates, indicating the importance of autophagy in constitutive surveillance of proteome quality in neurons. Additionally, mutations in several autophagy proteins have been linked to the progression of neurodegenerative diseases in humans, underscoring the essential role of autophagy in protecting neuronal health [209]. Maintaining the quality of the proteome and organelles is a point of vulnerability for neurons. In the aging brain, oxidative stress induces damage and modifications to DNA, lipids, and proteins, and the efficiency of the autophagic system decreases with age [210]. On the other hand, neurons have an elaborate morphology with highly branched dendritic networks and an axon that can reach up to a meter in length in humans. This complex architecture has a logistical challenge because, in the cell body, distant sites, such as the axons, the integrity of proteins, and organelles are hard to be maintained [211].

In this sense, autophagy likely plays a principal role in quality control at the synapse, where cellular components are overused and prone to damage. Another autophagy function implies neurotransmission regulation by controlling the number of synaptic vesicles. Therefore, the loss of autophagy affects several aspects of synaptic development, maintenance, and function. Collectively, autophagy performs diverse and compartment-specific functions in neurons. These functions can depend on the age of the neurons and can be specific to the type of neuron [209].

The glial cells account for ~90% of brain cells and are principal neuronal homeostasis and function regulators. Moreover, they influence many features of neuronal development, metabolism, synaptic function, and repair after injury. During stressful conditions, the glia can provide “help packages” to neurons in the form of exosomes containing mRNAs and translation machinery. Furthermore, alterations in neuronal–glial communication are crucial factors contributing to the progression of neurodegeneration. Recent evidence from *Caenorhabditis elegans* demonstrates that neurons release protein aggregates and organelles into membrane-bound vesicles. Subsequently, other cell types phagocytose them [212]. A report indicates that the glia exhibits higher proteasomal activity levels than neurons of the same age [213]. In Huntington’s disease, glia can effectively eliminate aggregates [209]. On the other hand, it has been observed in both animal and cellular models that autophagy contributes to different aspects of neuronal physiology, including axonal homeostasis, synaptic pruning, and neurogenesis, through the maintenance of neuronal progenitors [23,212,214]. During the postnatal period, several trophic factors, including BDNF, act in neuronal survival [215], since BDNF/TrkB signaling in neurons contributes to cell survival by activating the P13K/Akt/mTOR pathway and regulating autophagy [162].

## 4. Autophagy Modulator Factors

### 4.1. Trophic Factors and Autophagy

Neurotrophic factors (NTFs) are endogenous secreted proteins that promote the survival and differentiation of some neuronal populations, both in vivo and in vitro [216]. They regulate the formation and establishment of synaptic connections and gene expression through their interaction with specific cell receptors [217,218]. During the developmental PCD process, trophic factors play an important role in regulating the final number of neurons and connections in the nervous system. The same neuronal type can respond to several trophic factors, and a growth factor can affect different groups of neurons. In addition to acting endocrineally, trophic factors can also function through an autocrine mechanism. The same cell has receptors for growth factors on its plasma membrane, synthesizes trophic or paracrine factors, and secretes them close to the recipient cell. Finally, trophic factors have an anterograde mechanism of action. In this type of mechanism, the postsynaptic neuron receives trophic factors from the presynaptic neuron (Figure 5 and Figure 6) [219].

Neurotrophic factors are classified into at least three families based on their structures, receptors, and signaling pathways: (1) neurotrophins, including nerve growth factor (NGF), brain-derived neurotrophic factor (BDNF), neurotrophin-3 (NT-3), and neurotrophin-4 (NT-4/5); (2) transforming growth factor-β (TGF-β) superfamily, which includes the TGF-β family, the glial cell line-derived neurotrophic factor family (GDNF), and the BMP (Bone Morphogenetic Protein) family, and (3) family of neurocytokines [220], among which the CNTF (Ciliary Neurotrophic Factor) stands out [219,221].

In sum, neurotrophins are a family of proteins that regulate the generation, survival, proliferation, differentiation, and death of neurons in the central and peripheral nervous systems. They also facilitate synaptic transmission and plasticity, which are crucial for the memory and regeneration processes [222]. The factors of this neurotrophin family synthesize as 32 kDa precursor proteins called pro-neurotrophins, which undergo a subsequent proteolytic cleavage that shapes a mature constitution [223]. Mature neurotrophins bind to two transmembrane receptor classes: neurotrophin receptor p75 and tropomyosin receptor (Trk). The receptor p75 activation evokes cell death, and Trk activation allows cell survival [220].

At the cellular level, TGF-β superfamily proteins regulate fundamental processes: proliferation, differentiation, cell death, cytoskeletal organization, adhesion, and migration [224]. In humans, the TGF-β superfamily is composed of 33 members that can be subclassified into several subfamilies, including TGF-β (TGF-β1, -β2, and -β3), BMP, and GDNF [225]. BMPs are a family of proteins that control proliferation, differentiation, adhesion, motility, and survival in the nervous system. In addition, their great importance during embryonic development should be noted. They participate in developing the neural tube and establishing the dorsoventral axis at the mesodermal level. The effects of BMPs vary depending on exposure time, dose, and the presence of other signals or external factors [219].

On the other hand, members of the GDNF family (which also includes neurturin (NRTN), artemin (ARTN), and persephin (PSPN)) have neurotrophic effects on different neuronal populations of the CNS. GDNF can promote the survival of different types of neurons, such as dopaminergic, cholinergic of the basal nuclei, Purkinje of the cerebellum, and motor neurons. NRTN is a critical survival factor for parasympathetic neurons. Persephin promotes the survival of motor neurons and dopaminergic neurons of the CNS but does not exert effects on neurons of the PNS [219,226].

The neurocytokine family includes CNTF, interleukin-6, interleukin-11, leukemia inhibitory factor (LIF), oncostatin M, cardiotrophin-1, and granulocyte colony-stimulating factor. All members are related to cytokines and show analogy regarding the tertiary structure and signaling pathways through cell surface receptors. Neurocytokines are involved in neuronal and glial differentiation and development, regulate neuronal survival and phenotypic expression of neuropeptides and neurotransmitters, and rescue neurons from axotomy-induced cell death [221].

Additionally, fibroblast growth factor families (FGFs) form a large family of polypeptides characterized by their affinity to glycosaminoglycan heparin-binding sites [227,228]. The function of FGFs in developmental processes includes mesoderm induction, anterior-posterior patterning, limb development, and neural induction and development. Over 20 different FGF family members have been identified in mammals [229,230], all of which are structurally related signaling molecules. The most known members of this family are FGF1 (acidic FGF) and FGF2 (basic FGF), which have been studied extensively, and have mitogenic and survival properties in cells from different tissues, including the nervous system. Basic fibroblast growth factor (bFGF) has been reported to be highly expressed in the central nervous system (CNS) and to exert neuroprotective effects against different CNS diseases [231]. Similarly, VEGF (Vascular Endothelial growth factor) is another growth factor with essential survival functions, and it has been reported that its absence induces autophagy, such as the absence of bFGF, which also induces autophagy [232,233,234,235]. In general, autophagy is induced experimentally by drastic changes in culture conditions, such as fasting, absence of serum, absence of glucose, absence of amino acids, low levels of oxygen, and other types of cellular stress [236,237]. Accordingly, our group previously reported that autophagy could be induced by the removal of a single growth factor (bFGF) in neural precursor cells (NPCs). One of the most important results obtained in our work was the identification of a mechanism of cell death associated with the formation of vacuoles that occurs in dorsal midbrain NPCs cultured in the absence of bFGF. According to the morphological and biochemical data observed, it is a programmed process in which the de novo synthesis of RNA and proteins is required, similar to autophagy [235]. Although most studies associate autophagy with growth factors, such as IGF1 (factor insulin growth 1), other neurotrophic factors can also regulate it. For example, BDNF can improve neuronal survival through the positive regulation of autophagy in experimental systems. NGF can induce the autophagy process, although it is not yet clear how he does it. In sum, the relationship between trophic factors and autophagy is emerging as a relevant element that modulates neurodegenerative conditions [111].

Autophagy can be a protective mechanism for the organism in many respects, among them, because it is a slower process compared to apoptosis [238], which allows cells to remain alive for a longer time, waiting for better conditions in their microenvironment. Likewise, it could be a defense reaction to harmful stimuli, as has been seen to occur in MCF-7 cells (breast cancer cell line), which, when receiving low doses of radiation, show the characteristics of the autophagic process, which correlates with an increase in their survival capacity [239]. Other stimuli to induce autophagy apart from starvation are ionizing radiation [240], arsenic trioxide toxicity (a drug used to treat leukemia [241], the concentration of neurotransmitters such as dopamine [242], treatment with tamoxifen [243], endostatin (angiogenesis inhibitor [244] among others. In all these cases, the presence of autophagy could be a response of the cell to counteract stress, but if the harmful stimulus continues sooner or later, the cell will die. There is a demonstrative case of the protective effect of autophagy in a colon cancer cell line, in which when the autophagic process is inhibited, the sensitivity of these cells to apoptotic stimuli increases, so that autophagy, in this case, could represent an attempt of the cells to recover from the noxious stimulus, probably by sequestering the factors that promote death by apoptosis, such as cytochrome c, in the autophagic vacuoles [245], or simply by degrading only the organelles.

Autophagy is active at basal levels in most cells and is considered to play a role in maintaining intracellular integrity and homeostasis [30]. This recycling pathway is crucial in the adaptive response of cells and organisms to nutrient deprivation, which promotes survival until they become available again [246]. Autophagy is quickly regulated when cells need to generate nutrients and intracellular energy. For instance, during starvation, withdrawal from trophic factors, or high bioenergetic demands. This is because nutritional status, hormonal factors, and other signals, such as temperature, oxygen concentrations, and cell density, are essential for cell survival and homeostasis [246,247,248].

The rapamycin kinase (mTOR in mammals) and AMP-dependent protein kinase (AMPK) targets are nutrient sensors and play a principal role in autophagy regulation. They can detect nitrogen and carbon, respectively [28,247]. Both sensors regulate the ULK1-ULK2 complex through a series of phosphorylation events. For example, in a medium packed with amino acids, growth factors, and high ATP levels, mTOR activates and inhibits autophagy by binding to the ULK1 complex and phosphorylating ATG13 and ULK1, thereby suppressing kinase activity ULK1 and, therefore, the autophagy process [30,246,247]. On the other hand, AMPK activation promotes autophagy through allosteric AMP binding and Thr172 phosphorylation. This mechanism activates ULK1 through phosphorylation of Ser317 and Ser77 under glucose deprivation or Ser555 under amino acid deprivation and mitophagy [248].

Another regulatory molecule that controls autophagy is eukaryotic initiation factor 2α (eIF2α), a transcriptional transactivator of autophagy genes that activate autophagy during nutrient depletion [246,247]. Therefore, autophagy constitutes a feedback loop in response to a lack of nutrients [17].

### 4.2. Other Modulators of Autophagy:

#### 4.2.1. Age, Glucose, Amino Acids

Various stresses induce autophagy, including nutrient deprivation, growth factor withdrawal, genotoxic stress, damaged proteins and organelles, aging, and metabolic changes. Aging correlates with reduced autophagy in diverse organisms [249]. Some studies have revealed that aged rats, primary human cells, and *Caenorhabditis elegans* have reduced lysosomal proteolysis compared to younger individuals. Aging is also accompanied by the reduced expression of several ATGs in *Drosophila* and rodent tissues. In humans, ATG5, ATG7, and Beclin-1 are downregulated during normal aging [250]. Autophagy declines with age, and autophagy impairment predisposes individuals to various age-related diseases, including neurodegeneration and arthritis. Aging correlates with reduced autophagy in diverse organisms [249,250]. Evidence supports autophagy as a critical regulator of lifespan. Many physiological and pharmacological interventions extend lifespans, such as caloric restriction and mTOR inhibition by rapamycin, which induce autophagy. Through autophagy, cells coordinate energy and building blocks demanded for vital processes, such as growth and proliferation, that require extracellular stimuli and carbon source availability, such as amino acids and glucose. Both glucose and amino acid signals converge on a unique molecular transducer of cellular needs: the mammalian target of rapamycin complex 1 (mTORC1) [251]. Autophagy, as a regulator of tissue microenvironment metabolism in yeast amino acids generated by autophagy, can be used to sustain new protein synthesis and maintain mitochondrial functions under nutrient deprivation. Amino acids generated in the liver by autophagy are used for gluconeogenesis to maintain systemic glycemia under starvation [252]. Amino acids have been shown to promote the translocation of mTORC1 in a Rag-dependent manner to the surface in the endomembrane compartment, where mTORC1 can find its Rheb activator. In addition, the presence of a trimeric Ragulator protein complex is essential for activating the mTORC1 pathway via amino acids (Figure 6) [253].

#### 4.2.2. ROS and Autophagy

Oxidative stress is an imbalance between oxidants and antioxidants in favor of oxidants, disrupting redox signaling [254]. Redox signaling is the transduction of signals in which integrative elements are electron transfer reactions involving free radicals or related species, redox-active metals (e.g., iron, copper, etc.), or reductive equivalents [255]. Many reports have demonstrated that redox signaling affects autophagic flux, resulting in its induction [256]. However, prolonged autophagy may result in the degradation of essential proteins and organelles, leading to cell death (Figure 6) [257].

Studies have shown multiple molecular pathways involved in the regulation of ROS and autophagy. For instance, excessive generation of ROS activates HIF-1, p53, FOXO3, and NRF2. These transcription factors then induce the transcription of some autophagy regulator genes, such as BNIP3 and NIX, TIGAR, LC3, BNIP3, and p62. The corresponding protein products induce autophagy [258,259,260,261,262], which can be considered post-transcriptional regulation. On the other hand, autophagy also regulates levels of ROS by pathways such as the chaperone-mediated autophagy (CMA) pathway [54], P62 delivery pathway [263], and the mitophagy pathway (Figure 6) [264].

#### 4.2.3. Infections and Autophagy

Autophagy is also a pathway to the degradation of microorganisms, but several pathogens take advantage of the autophagy machinery to survive and replicate. Specifically, xenophagy, a type of selective autophagy, senses intracellular microorganisms and physically targets them to autophagosomes for further degradation [265].

In bacterial infections, SQSTM1/p62-like receptors (SLRs) recognize ubiquitinated substrates, recruit membranes to autophagosomes, and interact with LC3 [266,267]. Studies have demonstrated the capacity to eliminate *S. pyogenes* via the autophagic pathway [268]. Other studies have shown that *S. typhimurium* is also eliminated by autophagy [269]. In *M. tuberculosis* infections, autophagy seems to contribute to the bacterium’s elimination [270]. On the other hand, certain bacteria inhibit autophagy by interfering directly with autophagy components. An example is *Legionella pneumophila*, a bacterium that inhibits delivery to lysosomes [271]. *S. flexneri* is another example of a bacterium that interferes with the components of autophagy. *Shigella* can manipulate the autophagic pathway by secreting factors; these factors, IcsB and IcsA, reduce the binding of ATG5 and recruit tectonin beta-propeller repeat-containing protein 1(TECPR1), a tethering factor involved in autophagosome maturation. [272,273]. Finally, *P. gingivalis* is a periodontal pathogen that activates autophagy after being internalized. Although autophagosomes sequester this bacterium, they evade the formation of autolysosomes, thereby preventing its degradation [274].

Autophagy also occurs during parasitic infections and exhibits specific characteristics. For instance, observations suggest that autophagy could be involved in eliminating the tachyzoites of *Toxoplasma gondii* [275]. On the contrary, *Trypanosome cruzi* can manipulate host-cell autophagy to control the infection within a host [276]. Finally, autophagy induction drives *Echinococcus granulosus* death [277].

Although autophagy aims at clearance, some viruses have evolved a variety of strategies to inhibit, escape, or manipulate multiple steps with the aim of its survival and propagation. These viruses suppress the autophagic process to avoid degradation [278] or use the autophagosome for replication [279]. Autophagy is divided into several processes: induction, nucleation of the phagophore, elongation, fusion, and degradation. However, viruses have generated strategies to escape or manipulate these autophagic processes to benefit their replication and propagation [280,281,282].

Regarding SARS-CoV-2, accumulating evidence suggests that SARS-CoV-2 infection may perturb the autophagic process. For instance, one study demonstrated that SARS-CoV-2 reduces glycolysis and protein translation by limiting the activation of AMPK/MTORC1 signaling [283]. On the other hand, a report showed ORF3a (SARS-CoV-2 protein) interacts with the HOPS component VPS39 to block the interaction of HOPS with autophagosomal STX17, as well as to prevent or avoid lysosomal function [284]. Thus, the evidence suggests that SARS-CoV-2 infection is related to the induction of incomplete autophagy (Figure 2 and Figure 6).

## 5. Implications of Autophagy in Human Diseases

Autophagy plays an essential role in various physiological and pathological processes, including cell survival, organism development, bioenergetic homeostasis, cell death, aging, immunity, and metabolism. If it is not adequately regulated, it will trigger or spread several pathologies. Accumulating evidence has highlighted the importance of autophagy in many human diseases, such as cancer, neurodegenerative diseases, metabolic disorders, immunity, and infection. Therefore, a growing number of studies have explored the relationship between autophagy and human disease. The defective cell dead clearance mechanism in autophagy is associated with inflammatory diseases and neurodegenerative diseases. There has been considerable interest in modulating autophagy as a potential target in clinical medicine [285]. The lysosomal degradation pathway of autophagy plays a crucial role in cell physiology, including adaptation to metabolic stress, removing dangerous loads, renewal during differentiation and development, and preventing disease and damage. However, in certain circumstances, autophagy can be detrimental through its effects in favor of survival, such as cancer progression, or possible effects promoting cell death [286]. Most neurodegenerative diseases are associated with the intracytoplasmic deposition of proteins prone to aggregation in neurons and mitochondrial dysfunction. The autophagy process can eliminate proteins and maintain cell homeostasis. Upregulation of autophagy has been shown to protect against neurodegeneration [287,288,289,290] (See Table 4).

In the case of muscular cells, there is evidence showing that in the autophagy process, lysosomal degradation contributes to protein degradation in denervated muscles. It has been shown that cathepsin L, a lysosomal protease, is a lysosomal protease upregulated during muscle atrophy [291]. Autophagy has been identified as a protective mechanism in normal cartilage against oxidative stress and other aging-related phenotypes. Its age-related loss is linked to cell death and osteoarthritis. Cartilage-specific autophagy deficiency has been shown to contribute to growth retardation. Furthermore, in the senile population, the decreased autophagy activity of osteocytes with aging is proposed as an underlying cause of bone loss [292].

Many types of cancer are characterized by upregulation of the autophagy process, which seems to promote survival and increase malignancy, such as lung cancer. Autophagy has been shown to play a critical role in initiating and growing tumors and their metastasis. Autophagy is also involved in the chemoresistance of tumor cells [293]. Autophagy contributes to the effective quality control of organelles and cytosolic proteins in hepatocytes. Consequently, autophagy disorders or malfunctions are associated with pathological symptoms and liver diseases, including viral hepatitis and hepatocellular carcinoma. In particular, an aging-dependent decline in autophagic activity frequently underlies the pathogenesis of liver disease [294]. Several studies have revealed that genetic alterations in the autophagic or lysosomal pathways cause damage to the pancreas. Alteration of genes encoding proteins that mediate the formation of autophagosomes, such as *ATG5* or *ATG7,* or lysosomal function (LAMP2), causes blockage or impairment of autophagy in the pancreas, resulting in severe acinar cell degeneration, exocrine atrophy of the pancreas, fibrosis, and inflammation. Autophagy is critical for maintaining pancreatic acinar cell homeostasis, and impaired autophagy also deregulates secretion in LAMP2-deficient acinar cells [295,296].

Autophagy plays a critical role in intestinal homeostasis. Genetic alterations can involve genes that control autophagy and dependent processes, such as susceptibility to Crohn’s disease, an inflammatory bowel disease. These susceptibility loci include a coding variant in ATG16L1 and polymorphisms in IRGM, which implicate a role for xenophagy in intestinal homeostasis and disease [297].

Autophagy reduces injury and preserves cardiac function during ischemia. It also reduces chronic ischemic remodeling and mediates cardiac adaptation to pressure overload by restricting the accumulation of misfolded proteins, mitochondrial dysfunction, and oxidative stress. However, massive activation of autophagy can be detrimental to the heart under certain stressful conditions, such as reperfusion injury [298,299]. It has been reported that severe acute respiratory syndrome coronavirus 2 (SARS-CoV-2) inhibits autophagy flux and causes the accumulation of autophagosomes [300], as previously discussed in the present manuscript.

## 6. Conclusions

Autophagy is a natural process in which the cells of the body clean out any damaged or unnecessary components. Autophagy is an adaptive response to stress conditions, allowing the survival of cells for a longer time in adverse environments. Although autophagy was initially described as a conserved inducible pathway for the general turnover of cytoplasmic components [237], it is now also considered a relevant pathway for cellular degeneration [236,326]. Autophagy has been observed during the development of different organisms, as discussed in the present review, particularly in regions where abundant cell death occurs [327,328,329,330,331], thus suggesting that it contributes to programmed cell death (PCD).

Therefore, autophagy and apoptosis are in some way mutually exclusive, but autophagy in the latter stages can converge into late apoptosis under harmful conditions. On the other hand, during development, where PCD appears to be mainly executed by apoptosis, it has been observed that reduction in caspase activity, by enzyme inhibition [332] or by null mutations in gene coding triggering caspases (caspase 9 and caspase 3) [333], does not prevent cell degeneration but causes the switch of PCD from apoptosis to autophagy.

Autophagy plays a pivotal role in embryonic development, and morphogenesis, maintaining the organism’s homeostasis and eliminating damaged proteins and organelles. The normal development of cells depends on the correct regulation of the balance of protein synthesis and degradation and the biogenesis and degradation of organelles. Proteasome-mediated degradation (ATP-dependent, intracellular protease) is responsible for most, but not all, protein recycling [334]; whereas, the recycling of organelles is mainly attributed to autophagy [335]. Autophagy occurs in eukaryotic cells, where organelles and other cellular components are sequestered in double-membrane vacuoles, fusing with lysosomes for degradation. Autophagy is not exclusive to multicellular organisms, as described in protozoa such as *Leishmania donovani* [336] and yeasts, such as *Saccharomyces cerevisiae*. It is a highly conserved process on the evolutionary scale, from unicellular organisms through metazoans from *Caenorhabditis elegans* and *Drosophila melanogaster* to mammalian cells, including human cells. In general, it is considered that if a mechanism is conserved throughout the phylogenetic scale, it is because it has a determining relevance for life; therefore, autophagy could confer “advantages” to cells that can undergo it. Moreover, it has been reported that autophagy allows greater resistance to fasting and is an essential process for cell remodeling during differentiation, metamorphosis, aging, and cell transformation, as well as for participating in the remodeling of complete organs. such as, for example, during the growth of the uterus in the gestation period and its atrophy after the birth of the offspring, or, during the development of the nervous system, by removing abnormal or excess cells [337].

The proteins that participate in regulating the autophagy mechanism are highly conserved on an evolutionary scale. The autophagic mechanism overlaps with the mechanism for targeting cellular components to the vacuole (or lysosome), Cvt. However, the main difference is that autophagy is a degradative pathway triggered under fasting conditions, while the Cvt pathway is biosynthetic and occurs under nutrient-rich conditions. The vacuoles generated in both mechanisms are double-membranes, but those of autophagy are larger [236]. The regulation of the phosphorylation of *Atg* genes, involved in autophagy, through TOR kinase could be the “switch” that regulates the transition from the Cvt pathway to the Autophagic pathway under fasting conditions.

Several factors can regulate autophagy: absence of nutrients, absence of growth factors, levels of glucose, amino acids, ROS, stress, aging, infections, etc. Therefore, autophagy impairment is characteristic of several pathologies of multiple organs, including the brain, muscle, bones, lungs, liver, pancreas, gut, and heart. Understanding autophagy pathway regulation will help develop new modulators for developmental disorders and chronic-degenerative diseases. Autophagy occurs in neurodegenerative diseases characterized by protein misfolding or conformational changes. This vacuolation mechanism contributes to the removal of abnormally folded proteins inside or outside the endoplasmic reticulum [338]. Abnormal protein folding appears to be what stimulates (triggers) autophagy. However, excessive autophagy can contribute to neurodegeneration, as in the case of Huntington’s and Alzheimer’s diseases, by altering the processing of mutant forms of Huntingtin and Amyloid β-protein precursor, respectively [339,340]. Further investigation of the role of autophagy in neurodegenerative diseases is needed to better understand its role in pathophysiology to allow the design of improved therapeutic approaches.

## Figures and Tables

**Figure 1 cells-11-02262-f001:**
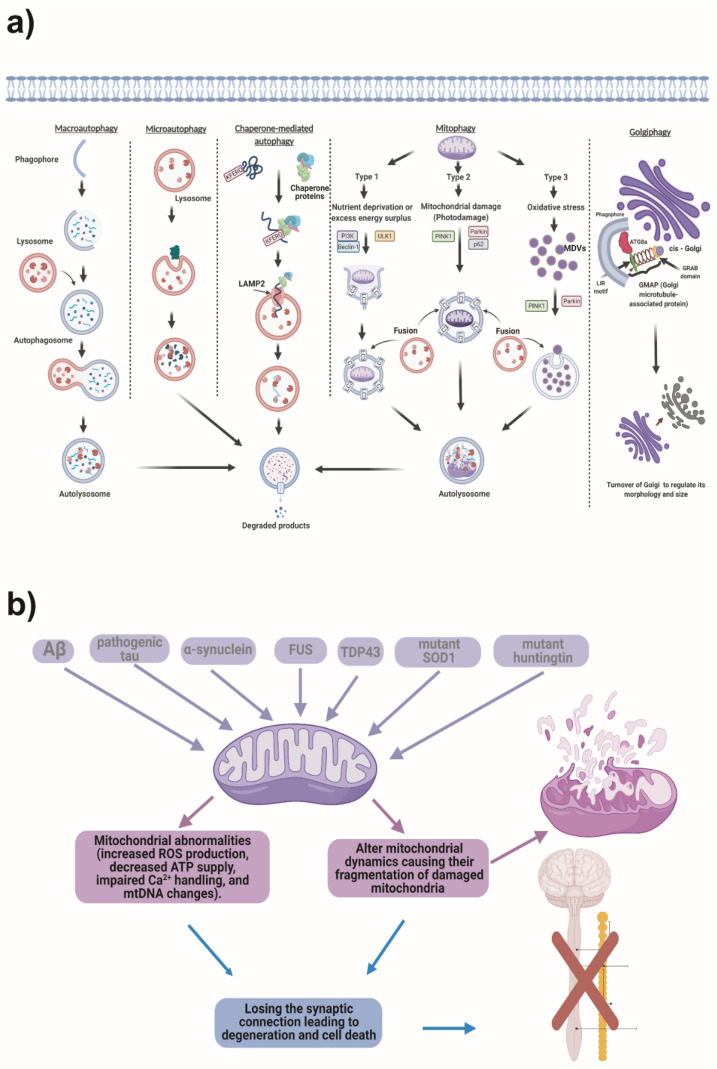
Autophagy types and mitochondrial stress. (**a**) Autophagy types: schematic representation of macroautophagy, microautophagy, chaperone-mediated autophagy, mitophagy, and golgiphagy. (**b**) Many factors can generate mitochondrial stress (Aβ, pathogenic tau, α-synuclein, FUS, TDP-43, mutant SOD1, and mutant huntingtin). The consequences are mitochondrial abnormalities and altered mitochondrial dynamics that lead to degeneration and cell death.“Created with BioRender.com (accessed on 1 May 2022)”.

**Figure 2 cells-11-02262-f002:**
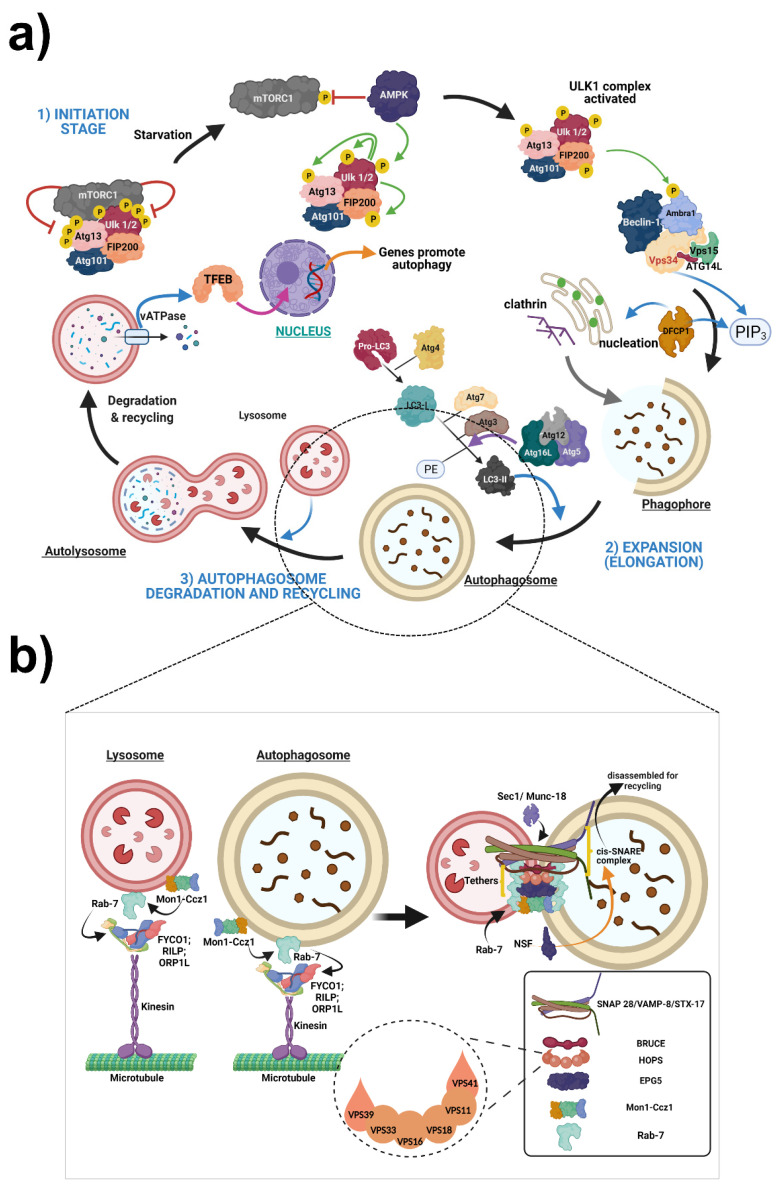
(**a**) Autophagy stage modulators. (1) Initiation stage: mTORC1 directly phosphorylates and inhibits the ULK-Atg13-FIP200 complex in nutrient-rich conditions. Phosphorylation of ULK1 and ATG13 prevents the induction of autophagy. Upon starvation, mTORC1 is inhibited and phosphorylated by AMPK, which results in ULK1 activation. Subsequently, the ULK1 complex phosphorylates the Ambra1 protein, which allows the recruitment of the Beclin-1-Vps34 complex (Beclin1-Vps34-Vps15-ATG14L). Vps34 produces phosphatidylinositol-3-phosphate (PIP3) and recruits DFCP1 to promote nucleation. Via endocytosis mediated by Clathrin generates the phagophore. (2) Expansion (Elongation): pro-LC3 is cleavaged by ATG4B and generates LC3-I. Then, ATG7 and ATG3 process LC3-I to be conjugated to PE and the ATG12-ATG5-ATG16L system to produce LC3-II. The formation of this complex is necessary for the elongation of phagophores. (3) Autophagosome Degradation and Recycling: Mature autophagosomes fuse with lysosomes to form autolysosomes. In lysosomes, degradation occurs with hydrolytic enzymes that are active in acidic pH. Vacuolar ATPase (vATPase) regulates this pH. Finally, degraded products are released into the cytosol to be recycled. On the other hand, vATPase induces transcription factor EB (TFEB) activity in starvation conditions. TFEB translocates to the nucleus, induces the transcription of genes that promote autophagy, and reinforces autophagy induction. (**b**) Mechanisms of fusion of autophagosome to Lysosome to allow autophagy flux. First, autophagosomes and lysosomes transport are essential in the fusion. In this sense, Mon1-Ccz1 recruits and activates Rab7. Rab7 binds to FYCO1, ORP1L, and RILP, mediating vesicle transport. Once autophagosomes and lysosomes reach their destination, multisubunit complexes (EPG5, HOPS complex, and BRUCE) tether them to their partner organelle. Then, they bind to Rab-7. Next, SM (Sec1/Munc-18) family proteins arrive and facilitate SNARE complex (STX17-SNAP29-VAMP8) assembly and zippering after all SNAREs are located on the same membrane. The cis-SNARE complex is then recognized and disassembled by NSF for recycling “Created with BioRender.com (accessed on 1 May 2022)”.

**Figure 3 cells-11-02262-f003:**
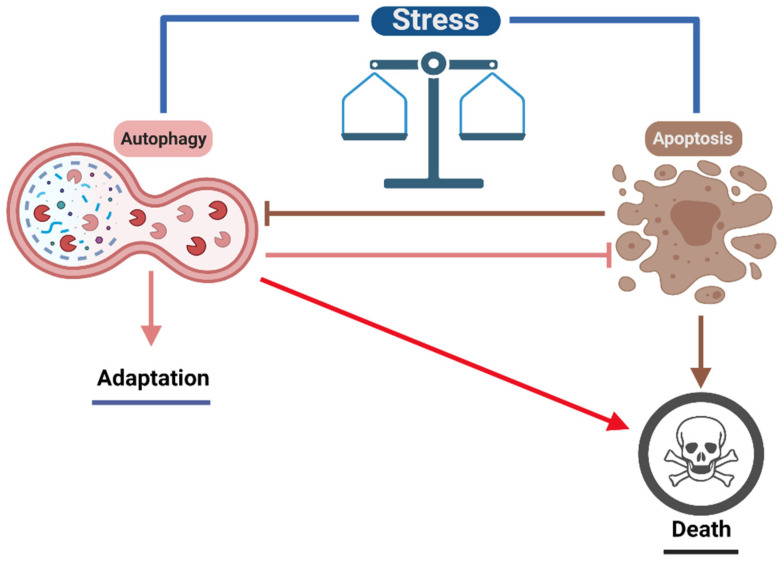
The balance between autophagy and apoptosis in stress conditions. According to stress stimuli, the cell has an adaption process associated with autophagy or decides to die in an apoptosis process. The choice depends on the microinvironmetal: nutrient availability, and prolonged time in stress conditions, among others. “Created with BioRender.com (accessed on 1 May 2022)”.

**Figure 4 cells-11-02262-f004:**
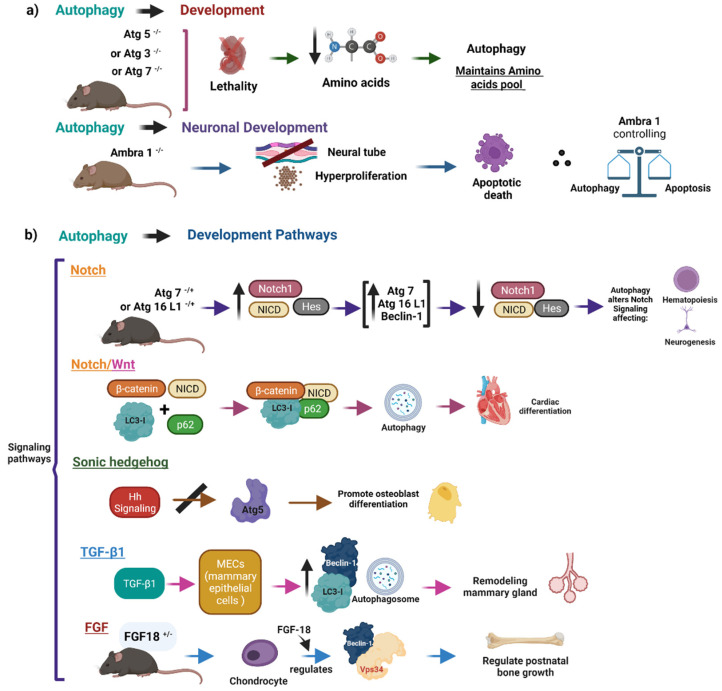
The role of autophagy in the regulation of embryonic development. (**a**) Autophagy as a regulator of embryonic development and neural development. Mouse models suggest that autophagy is essential for development. (**b**) Autophagy interacts with developmental signaling pathways, among them Notch, Wnt, Sonic Hedgehog, FGF, and TGFβ. Impairments in the crosstalk between these pathways and autophagy could likely be the cause of the defects in the generation of embryonic structures. See the text above for more details. “Created with BioRender.com (accessed on 1 May 2022)”.

**Figure 5 cells-11-02262-f005:**
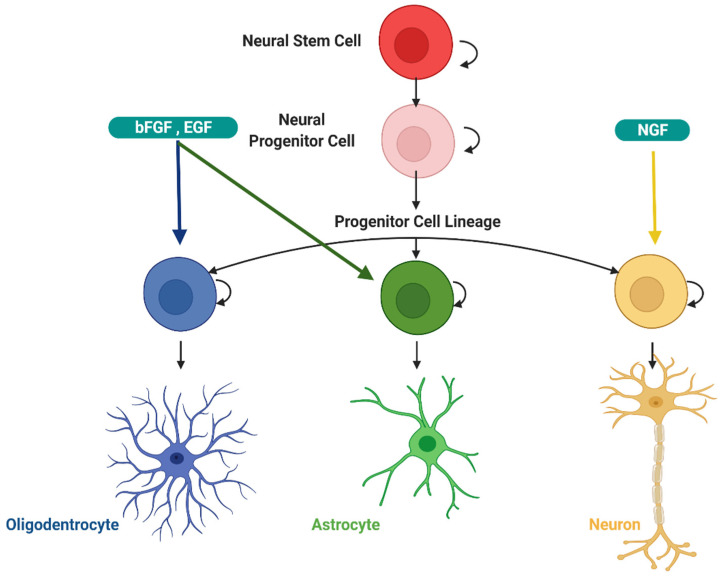
Neurogenesis and trophic factors. Neural stem cells generate progenitor cells. Trophic factors (bFGF and EGF) are necessary for the survival and proliferation of neural progenitor cells, while neurotrophin NGF promotes neuronal differentiation. Neural progenitors can give rise to the three principal phenotypes of the CNS: Neurons, Astrocytes, and Oligodendrocytes. “Created with BioRender.com (accessed on 1 May 2022)”.

**Figure 6 cells-11-02262-f006:**
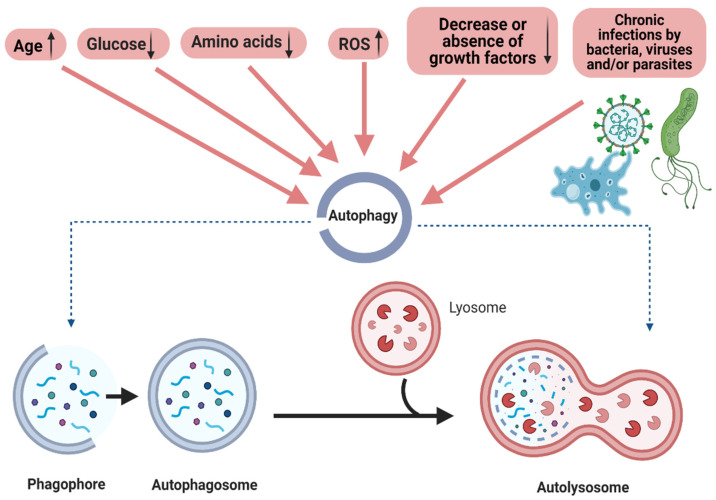
Modulator factors for autophagy. Several factors regulate autophagy. For instance, aging, glucose, and amino acids decreased, growth factor withdrawal, ROS, stress increased, and chronic infections by bacteria, viruses, or parasites “Created with BioRender.com (accessed on 1 May 2022)”.

**Table 1 cells-11-02262-t001:** Autophagy physiological functions, main characteristics.

Physiological Functions of Autophagy	Description	References
Energy Production	Autophagy can contribute to the mobilization of diverse energy molecular storages, regulating glucose metabolism, mobilization of neutral lipids, or is responsible for intracellular protein breakdown.	[5,6,7]
Intracellular Maintenance	Autophagy can contribute to the quality control and replacement of cellular components.	[8,9,10]
Signaling	Autophagy could, in some cases, distinguish its cargos, leaving the other cellular components unaffected because they are labeled with poly-Ub chains for degradation, a level of regulation associated with signaling.	[11,12,13]
Development and Differentiation	Autophagy can drive the rapid response of cells necessary for the development and play an indispensable role in differentiation. It provides the cells with the energy needed for differentiation and participates in tissue remodeling.	[14,15,16]

**Table 2 cells-11-02262-t002:** Autophagy types and mechanisms involved.

Autophagy Types	Mechanism	References
Macroautophagy	The isolation membrane sequesters cytosolic material to generate the autophagosome, which binds to the lysosome giving rise to the autolysosome, where cargo is degraded.	[24,76,77]
Microautophagy	The cytoplasmic material directly enters the lysosomal through invaginations of the lysosomal membrane.	[22,38,78]
Chaperone Mediated Autophagy	Labeled proteins with the pentapeptide KFERQ are recognized by Hsc70 and other chaperones that unfold them and then translocate them to the lysosomal lumen throw LAMP2.	[42,79,80]
Mitophagy	Dysfunctional or damaged mitochondria are degraded through the Parkin-dependent and independent pathways.	[81,82,83]
Golgiphagy	Golgiphagy is a selective type of autophagy, that regulates Golgi Complex Turnover. The GMAP (Golgi microtubule-associated protein) protein interacts with Atg8a and the LIR motif at position 320P-325, which is important for docking between the phagophore and the Golgi complex.	[75]

**Table 3 cells-11-02262-t003:** Role of autophagy proteins.

Stage	Yeast Proteins	Mammalian Homologous	Regulatory Function in Autophagy	Interaction
Induction	ATG1	ULK1, ULK2	Interacts with mTORC1; necessary initiation of autophagy regulation	ATG13, ATG1, ATG17
	ATG13	ATG13	Control autophagy induction modulating enzyme activity and cellular localization of ULK1	ULK1, ULK2, FIP200
	ATG17	FIP-200	Autophagy initiator	ULK1, ATG13, ATG101
Nucleation	ATG6	Beclin 1	Bcl-2 binding protein creates a regulatory complex with PI3K class III (VPS34).	Vps34, Pl3K, UVRAG
	ATG9	ATG9A	Associates to the pre-autophagosome structure. In yeast, it helps assemble the autophagosome.	ATG2
		ATG9B		
	ATG14	ATG14L	Autophagy specific subunit from the complex of PI3K Class III and Beclin-1	LC3
	ATG18	WIPI1-4	It binds to PI3P, possible participation in the nucleation stage.	DFCP1
Elongation	ATG3	ATG3	Ubiquitin E2 like enzyme acts as ligase of ATG8 and ATG12 and catalyzes the conjugation of ATG8 similar proteins to phosphatidylethanolamine (PE).	ATG7, ATG8, ATG12
	ATG4	ATG4A-D	ATG8 cysteine peptidase converts pro-LC3 (ATG8) into LC3-I and delipidated autophagosomal LC3-II.	ATG8
	ATG5	ATG5	ATG5 makes a complex with ATG12 and helps with the autophagosome elongation.	ATG12, ATG16L
	ATG7	ATG7	Ubiquitin E1 conjugase-like enzyme helps conjugate ATG8 to PE and acts as E1 enzyme for the conjugation of ATG12 to ATG5 and ATG3.	ATG8, ATG3, ATG12
	ATG8	LC3A,	Ubiquitin-like modifier; it associates with a stable component for the autophagosomal membrane	ATG3, ATG4, ATG7
		LC3B,		
		LC3C		
	ATG10	ATG10	Ubiquitin E2 type enzyme catalyzes the conjugation of ATG5 and ATG12.	ATG12, ATG16L
	ATG12	ATG12	Complex with ATG5 and helps in the autophagosomal elongation.	ATG3, ATG5, ATG7, ATG10, ATG16
	ATG16	ATG16L1/L2	Associated with the isolation membrane, making a complex with Atg5-Atg12, helps in autophagosomal elongation.	ATG5, ATG12

Based on the information of [2,30,113,114].

**Table 4 cells-11-02262-t004:** Diseases related to autophagy.

Role of Autophagy in Diseases			
Organ	Diseases	Autophagy Function	References
whole body (general function)	Tumor suppression and progression.	Selective degradation of p62, damaged mitochondria, and microbes; starvation-induced amino acid production; recycling of cytoplasmic content.	[4,301,302,303]
Brain	AD, amyotrophic lateral sclerosis (ALS), frontotemporal dementia, HD, and PD.	Prevention in the formation of aggregates, Parkin-dependent mitophagy, nutrient regulation, and energy balance.	[30,287,301,304,305]
Muscle	Danon disease, X-linked myopathy (XMEA), Ullrich congenital muscular dystrophy, Bethlem myopathy, sarcopenia.	Maintains muscle mass.	[291,306,307,308]
Bone	Paget’s disease, osteopetrosis, and osteopenia.	Bone metabolism. Terminal differentiation of osteoblasts. Maintains bone mass.	[292,309,310,311]
Lung	Cystic fibrosis, pulmonary tuberculosis, pulmonary arterial hypertension, and lung cancer.	Regulation of the responsiveness of the airways. Drive and regulate inflammatory responses in chronic lung diseases.	[293,312,313,314]
Liver	Hepatocellular carcinoma, alcoholic ad nonalcoholic fatty liver disease, α-antitrypsin deficiency. Viral hepatitis.	Adaptation to starvation through induction of glycogenolysis, lipolysis and protein catabolism, prevention of hepatocellular degeneration, and suppression of liver tumors.	[294,315,316,317]
Pancreas	Pancreatitis, and diabetes.	Adaptation of B cells to a diet rich in fat; prevention of trypsin autoactivation.	[295,296,301,318]
Gut	Crohn’s disease.	Maintenance of Paneth cell function.	[297,319,320,321]
Heart	Heart failure and atherosclerosis.	Adaptation to hemodynamic stress; prevention of age-dependent dysfunction.	[4,298,299,322]
Multi-Organ	COVID-19	Inhibition of the Autophagy flux by SARS-CoV-2.	[300,323,324,325]

## Data Availability

Not applicable.

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
