# Peer review of "Autophagy: A Key Regulator of Homeostasis and Disease: An Overview of Molecular Mechanisms and Modulators"

_cells, 2022, doi:10.3390/cells11152262_

Round 1

Reviewer 1 Report

The review titled "An Overview of the Molecular Mechanisms and Modulators of the Autophagy Pathway During Development and Cell Homeostasis" by Gómez-Virgilio et al. discuss the autophagy  in relation to cell death.

The role of autophagy can be controversial in different context.

Only minor point.

The references can be improved, the authors does not cite for example the last edition of the Guidelines for the use and interpretation of assay for monitoring autophagy  by Klyonski. 2021

Similarly, the autophagy has been shown to protect against neurodegeneration several other references can be added (Fornai et al. Autophagy  2008; Forani et al 2008 PNAs; Natale et al 2015)

In lines 816-824 there are repeated sentences. 

Author Response

Reviewer 1.

Answer to Reviewer 1:

Thank you for your valuable observations of our manuscript.

-The references can be improved, the authors does not cite for example the last edition of the Guidelines for the use and interpretation of assay for monitoring autophagy by Klyonski. 2021

Answer: We have now included Klyonski 2021 reference of the Guidelines for the use and interpretation of assay for monitoring autophagy, in two sections of our manuscript: In section 2.1. Brief summary of Key Autophagy players, of the introduction (Page 4, Line 100) and in section: Autophagosome Degradation and Recycling (Page 17, line 571). Please see the modifications labeled with the Word control of changes on the manuscript.

-Similarly, the autophagy has been shown to protect against neurodegeneration several other references can be added (Fornai et al. Autophagy  2008; Forani et al 2008 PNAs; Natale et al 2015)

Answer: We have included the three references that you suggested for the topic of autophagy function to protect against neurodegeneration. (Page 30, line 1130).

-In lines 816-824 there are repeated sentences. 

We have erased the duplicated sentence. Thank you for pointing out this duplication.

Note: We have reviewed the whole manuscript, introduced an index at the beginning, and reordered some of the sections to make it easy to follow by the readers, as suggested by Reviewer number 2. We have edited one of our figures to include the newly described type of autophagy, called Golgiphagy (Figure1) and briefly described in the text, and include a new figure (Figure 4), that describes the key role of autophagy on development. We also included a paragraph about this point of the relevant pathways regulating the role of autophagy on development. And review the grammar of all the manuscript and all changes can be identified by the Word control of changes.

Finally, we prepared a graphical abstract for our Manuscript.

We are resubmitting the revised version of our manuscript.

Reviewer 2 Report

Gomez-Virgilio et al. have written a comprehensive review on autophagy and its role in cellular homeostasis. The different types of autophagy and the molecular mechanisms have been covered. The tables and figures are great reference points and will be very useful in helping the reader to find important information quickly. However, due to the comprehensiveness and possibly also due to lack of insight in the autophagy field, the manuscript loses its focus, becomes disorganized and lacks a logical and structured flow to guide the reader. In fact, the manuscript loses track and does not reflect the title with regards to “development” and “cell homeostasis”. Many molecular terms are introduced early in the manuscript (p.3-6) yet the autophagic processes are not delineated in a structured fashion. For example, p62 – a prominent cargo receptor – is not mentioned until p.12. A similar scenario is seen for key autophagy players Beclin and LC3. Furthermore, large passages of text are repeated, for example, passage on autophagy on p.19, “ROS and autophagy” on p.20 and “implications of autophagy in human diseases” on p.24. Why does the section on apoptosis on p.23 come so late after the first mentioned of programmed cell death on p.16? Why are there two abbreviations used for programmed cell death (MCP for neurons and PCD in the conclusions)?

The authors are urged to perform major restructuring of the manuscript, such that the readers are able to follow their lines of thought. Irrelevant referencing to other diseases (Alzheimer’s, breast cancer) should be limited to only when absolutely necessary. The section on development, such as molecular details to the impact of autophagy on Notch signaling, for example, could be expanded to match the level of molecular details given in other sections. Is this only relevant for neural cells and blastocysts? What is the relevance of autophagy during organ development? How are other developmental signaling pathways, such as Shh, Wnt, TGFbeta, FGF, impacted? Are these signaling proteins degraded by autophagy? Finally, the introductory part should be making a clear statement on whether autophagy is physiological or not.

Author Response

Answers to Reviewer 2:

-Gomez-Virgilio et al. have written a comprehensive review on autophagy and its role in cellular homeostasis. The different types of autophagy and the molecular mechanisms have been covered. The tables and figures are great reference points and will be very useful in helping the reader to find important information quickly. However, due to the comprehensiveness and possibly also due to lack of insight in the autophagy field, the manuscript loses its focus, becomes disorganized and lacks a logical and structured flow to guide the reader. In fact, the manuscript loses track and does not reflect the title with regards to “development” and “cell homeostasis”. Many molecular terms are introduced early in the manuscript (p.3-6) yet the autophagic processes are not delineated in a structured fashion. For example, p62 – a prominent cargo receptor – is not mentioned until p.12. A similar scenario is seen for key autophagy players Beclin and LC3. Furthermore, large passages of text are repeated, for example, passage on autophagy on p.19, “ROS and autophagy” on p.20 and “implications of autophagy in human diseases” on p.24. Why does the section on apoptosis on p.23 come so late after the first mentioned of programmed cell death on p.16? Why are there two abbreviations used for programmed cell death (MCP for neurons and PCD in the conclusions)?

-The authors are urged to perform major restructuring of the manuscript, such that the readers are able to follow their lines of thought. Irrelevant referencing to other diseases (Alzheimer’s, breast cancer) should be limited to only when absolutely necessary. The section on development, such as molecular details to the impact of autophagy on Notch signaling, for example, could be expanded to match the level of molecular details given in other sections. Is this only relevant for neural cells and blastocysts? What is the relevance of autophagy during organ development? How are other developmental signaling pathways, such as Shh, Wnt, TGFbeta, FGF, impacted? Are these signaling proteins degraded by autophagy? Finally, the introductory part should be making a clear statement on whether autophagy is physiological or not.

We thank your valuable observations and we considered all of your recommendations and edited our manuscript accordingly:

We have reviewed the whole manuscript, introduced an index at the beginning, and reordered and organized some of the sections to make it easy to follow by the readers, as suggested by you. We have edited one of our figures to include the newly described type of autophagy, called Golgiphagy (Figure1) and briefly described in the text and table 1, and include a new figure (Figure 4), that describes the key role of autophagy on development. We also included a paragraph about this point of the relevant pathways regulating the role of autophagy on development (Shh, Wnt, TGFbeta, FGF). And review the grammar of all the manuscript and all changes can be identified by the Word control of changes.

We included several new key references of several aspects of autophagy.

Finally, we prepared a graphical abstract for our Manuscript.

We are resubmitting the revised version of our manuscript.

Round 2

Reviewer 2 Report

The manuscript by Gomez-Virgilio et al. has been vastly improved by their changes. The structure is logical and presentation of the subject matter is given in the right sequence. A few minor points need to be addressed:

The purpose of Figure 1B is not clear. It is also not cited in the main text.

Figure 4 is somewhat oversimplified and a little misleading since some signaling pathways target autophagy whereas others are affected by autophagy. Please modify according to the text. A figure legend is also missing.

The title still does not reflect the subject matter of the manuscript (development is one part but modulators and other diseases make up significant parts). Please edit the title to reflect the contents of the manuscript.

Author Response

Point by Point Answers to Reviewer 2:

-The manuscript by Gomez-Virgilio et al. has been vastly improved by their changes. The structure is logical and presentation of the subject matter is given in the right sequence. A few minor points need to be addressed:

Answer: Thank you for your commentary, we have improved the structure and sequence of our review following your recommendations.

-The purpose of Figure 1B is not clear. It is also not cited in the main text.

Answer: We have included a paragraph regarding Figure 1b, with two new citations at the end of the Mitophagy topic, and this figure was also cited in section: 2.3. Mitochondrial Stress and Autophagy.

-Figure 4 is somewhat oversimplified and a little misleading since some signaling pathways target autophagy whereas others are affected by autophagy. Please modify according to the text. A figure legend is also missing.

Answer: We have designed a more elaborated Figure 4, and modified it accordingly to the text of our manuscript. And we had included a legend for this figure.

- The title still does not reflect the subject matter of the manuscript (development is one part but modulators and other diseases make up significant parts). Please edit the title to reflect the contents of the manuscript.

Answer: Thank you for your suggestion, we have generated a new title to our review that reflects better the contents of our manuscript:

“Autophagy: a key regulator of homeostasis and disease. An overview of molecular mechanisms and modulators”.

Note: We have also reviewed the grammar of the whole manuscript and added extra information to the author affiliations, funding, and author contribution sections.

We are also resubmitting the New Revised Version of our Manuscript, including the new Figure 4.

Thank you very much for all your observations that helped us to improve our manuscript.

Kind regards

Dr. Maria-del-Carmen Cardenas-Aguayo

Professor

Department of Physiology,

School of Medicine, UNAM